

# HIRHAM–NAOSIM 2.0: The upgraded version of the coupled regional atmosphere-ocean-sea ice model for Arctic climate studies

Wolfgang Dorn[1], Annette Rinke[1], Cornelia Köberle[2], Klaus Dethloff[1], and Rüdiger Gerdes[2,3]

[1]Alfred Wegener Institute, Helmholtz Centre for Polar and Marine Research, Potsdam, Germany
[2]Alfred Wegener Institute, Helmholtz Centre for Polar and Marine Research, Bremerhaven, Germany
[3]Jacobs University, Physics and Earth Sciences, Bremen, Germany

**Correspondence:** Wolfgang Dorn (wolfgang.dorn@awi.de)

**Abstract.** A new version of the coupled Arctic atmosphere-ocean-sea ice model HIRHAM–NAOSIM is described. This version utilizes upgraded model components for the coupled subsystems, which include physical and numerical improvements and higher horizontal and vertical resolution, and a revised coupling procedure with the aid of the coupling software YAC. The model performance is evaluated against observationally based data sets and compared with the previous version. Ensemble

simulations for the period 1979–2016 reveal that Arctic sea ice is thicker in all seasons and closer to observations than in the previous version. Wintertime biases in sea-ice extent and near-surface air temperatures are reduced, while summertime biases are of similar magnitude as in the previous version. Problematic issues of the current model configuration and potential corrective measures and further developments are discussed.

## 1   Introduction

Coupled regional atmosphere-ocean-sea ice models allow to simulate interactions and feedbacks between its components which often occur at regional fine spatial and temporal scales that are not resolved by global models. Such feedbacks between atmosphere, sea ice, and ocean play a critical role in the evolution of the Arctic climate system and are likely to be jointly responsible for the climate phenomenon called Arctic Amplification (Kirtman et al., 2013). A variety of such feedbacks exist from which many are not yet understood in detail. Since these feedbacks include sub-processes, like cloud formation or boundary layer

turbulence, which operate below the model resolution and need to be parameterized on the basis of a process understanding, they are not sufficiently represented in climate models, indicating that model upgrades are still needed.

A coupled regional climate model can be an especially useful tool for providing information on the mechanisms behind the feedbacks between atmosphere, sea ice, and ocean as well as for testing different parameterizations of processes involved in these feedbacks. The regionally limited model domain with its constrained lateral boundary conditions is a major advantage in

this regard, since the model does not need to show reasonable performance in simulating global teleconnections. For instance, the simulation of the Atlantic Meridional Overturning Circulation (AMOC), which can sometimes be problematic in global circulation models (Danabasoglu et al., 2014; Gent, 2018), is not a major issue for a regional model with prescribed boundary conditions.



Here, we describe the upgraded version of the coupled Arctic atmosphere-ocean-sea ice model HIRHAM–NAOSIM (HN2.0) that succeeds an earlier version (HN1.2) but utilizes upgraded components, which include physical and numerical improvements and higher horizontal and vertical resolution, and a completely revised coupling procedure. The first version of HIRHAM–NAOSIM (HN1.0) was described by Rinke et al. (2003). Subsequent minor upgrades of the model (henceforth

referred to as HN1.1 and HN1.2) were described by Dorn et al. (2007) and Dorn et al. (2009). The extensive present upgrade of the coupled model was performed in the framework of the "ArctiC Amplification: Climate Relevant Atmospheric and SurfaCe Processes, and Feedback Mechanisms (AC)[3]" project with the aim to investigate interactions between atmosphere and sea ice in the Arctic (Wendisch et al., 2017). Details to the model components and to the coupling are given in section 2, while a basic evaluation of the current "base" configuration of HN2.0 is given in section 3. Major improvements as compared to the

previous version (HN1.2) as well as known weaknesses and potential further development of the model are discussed and summarized in section 4. It should be noted that the current base configuration is designed to serve as a reference for contemplated improvements of interaction processes between atmosphere, sea ice, and ocean.

## 2    Description of HN2.0

HIRHAM–NAOSIM 2.0 consists of the regional atmospheric climate model HIRHAM5 and the regional ocean-sea ice model

NAOSIM. Compared to the earlier versions (HN1.x), the atmospheric component uses both a new dynamical core and a new set of physical parameterizations, while the ocean-sea ice component is largely the same with only a few physical, but several technical improvements. Since the dynamical cores and the physical parameterizations of the two components were already explicitly described in scientific documentations or respective reference manuals, the focus here will be to explain the adaptations made in HN2.0 with respect to the original formulations of the model components and to detail the coupling

between HIRHAM5 and NAOSIM. Modifications of physical parameterizations, motivated to improve the model performance in the Arctic, will be explained as well.

### 2.1    Atmosphere model HIRHAM5

#### 2.1.1    Components and modifications

The regional atmospheric climate model HIRHAM5 consists of the hydrostatic dynamical core from the time integration

module of the weather forecast system HIRLAM-7.0 (Undén et al., 2002) and the physical parameterizations of the atmospheric general circulation model ECHAM-5.4.00 (Roeckner et al., 2003). The communication between the HIRLAM and ECHAM5 routines is handled by a separate interface, whereby most of the original model code could be adopted as it stands.

Modifications in the HIRLAM code are related to the data management, especially with respect to input and output data and restart functionality as detailed by Christensen et al. (2007), and to the separation of cloud water into liquid water and ice. The

latter is required by the ECHAM5 physical parameterizations and was realized by writing cloud liquid water on HIRLAM's total cloud water variable and cloud ice on HIRLAM's first optional tracer variable. The code has been consequently modified





such that the first tracer variable is treated in exactly the same way as the original total cloud water variable. The bottom line is that HIRHAM5 solves prognostic equations for seven instead of six variables, namely surface pressure, two horizontal wind components, temperature, specific humidity, cloud water, and cloud ice.

Modifications in the ECHAM5 code are kept to a minimum, but could not fully be avoided. Due to different grid definitions in the regional model (rotated spherical grid) as compared to the global ECHAM5 model (Gaussian grid), adaptations were necessary with respect to the initialization of radiation, ozone, aerosole, and soil parameters. Furthermore, a few parameterizations were modified to improve the model performance in the Arctic.

The most important modification concerns the surface albedo parameterization and was carried over from HN1.2 for the most part. The original snow albedo for non-forested land areas was replaced by the polynomial temperature dependency derived by Roesch (1999). For the sea-ice albedo, suggestions by Køltzow (2007), which particularly include the effect of melt ponds, were implemented. Version 1 of the sea-ice albedo scheme of Køltzow (2007) was implemented for HIRHAM5 stand-alone simulations, while version 2 was implemented for coupled HIRHAM–NAOSIM simulations. The main difference between version 1 and version 2 is the additional consideration of a snow cover fraction in version 2 which is derived in coupled mode from NAOSIM's prognostically computed snow thickness. A detailed description of the sea-ice albedo scheme in HIRHAM–NAOSIM was already given by Dorn et al. (2009). HN2.0 includes two modification: First, the linear transition towards the water albedo for thin ice (see Dorn et al., 2009, their equation (33)) is not yet activated; second, the restriction of the melt pond fraction to the sea-ice surface not covered with snow (see Dorn et al., 2009, their equation (37)) was attenuated such that at least 25 % of the computed melt pond fraction is allowed. The second modification was done to overcome too low sea-ice melt rates in early summer when the incoming shortwave radiation is at the maximum and a high amount of snow is still present. The value of 25 % can be considered as a tuning value. A more realistic parameterization of the fractions of snow and melt ponds should be derived from observations. This will be subject of future development (see also section 4).

The second modification concerns the stratiform cloud parameterization and includes a more efficient Bergeron-Findeisen process and a more generalized subgrid-scale variability of total water content. The main aim was to improve the simulation of low stratiform mixed-phase clouds, since these clouds occur frequently during all seasons in the Arctic (Morrison et al., 2012). A detailed description of the modified cloud parameterization is given by Klaus et al. (2016) who showed that this modification leads to significant improvements in the simulation of Arctic clouds and the surface radiation budget in HIRHAM5. This modification was implemented as an option and can be switched on or off.

The third and last modification concerns the calculation of the sea-ice temperature and takes only effect in coupled mode with NAOSIM. In that case, the sea-ice temperature is calculated from the surface energy balance according to the description of the ice growth parameterization by Dorn et al. (2009). The key change to the original ECHAM5 parameterization relates to the heat capacity of the uppermost snow/ice layer (see Dorn et al., 2009, their equations (21) to (23)). The computation of the heat capacity is substantially the same as in HN1.2, but $\delta h_0 = 0.05$ m is set as the maximum thickness of the uppermost snow/ice layer, motivated by the longer time step in HN2.0 as detailed below.





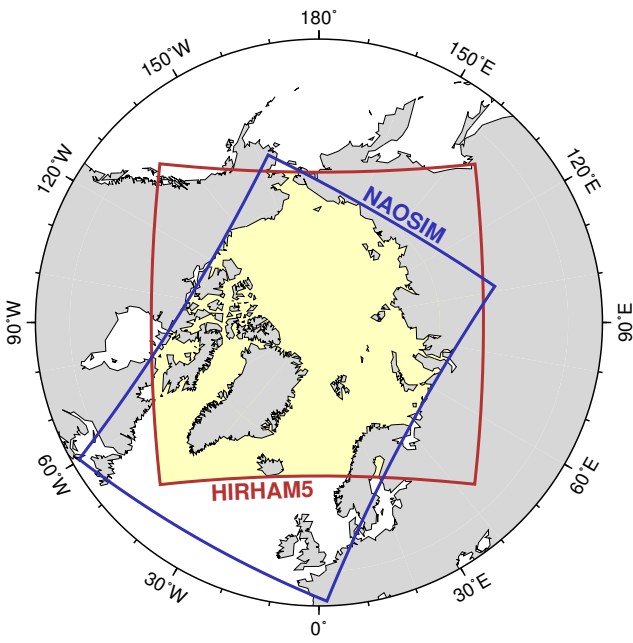

**Figure 1.** Geographical location of the regional model domains of the atmosphere component HIRHAM5 and of the ocean–ice component NAOSIM. The coupling domain is indicated by the yellow area and comprises 20583 (out of 43600) HIRHAM5 grid cells and 200951 (out of 366850) NAOSIM grid cells.

### 2.1.2 Domain and discretization

The model domain of HIRHAM5 covers the whole Arctic north of about 60° N (see Figure 1) and is defined in a rotated spherical coordinate system with the North Pole on the geographical equator at 180° E. The longitude range in the rotated pole grid is from 27.375° W to 26.875° E, and the latitude range is from 25.125° S to 24.625° N. The horizontal resolution is 0.25°

(approximately 27 km), resulting in a horizontal grid with 218×200 grid points. The dynamical core uses an Arakawa C-grid for the horizontal discretization. This means that the horizontal wind components are staggered in the horizontal with respect to the other prognostic variables that are discretized on the grid given above. HIRLAM internally handles the staggering and de-staggering of the horizontal wind components so that finally all variables are defined at the same grid points.

The vertical discretization is given in hybrid terrain-following/pressure coordinates with 40 levels from about 10 m above the

surface up to a pressure level of 10 hPa. The highest vertical resolution is in the lower troposphere, where the lowermost 1 km is represented by 10 levels (see Table A1). A 60-level version, with 20 additional levels in the stratosphere, is also available, but currently not used in coupled mode.

In addition to the traditional Eulerian time stepping scheme, HIRLAM-7.0 includes a semi-implicit semi-Lagrangian scheme. The semi-Lagrangian scheme allows for longer time steps and is now the default. Simulations presented in this paper were

carried out with a time step of 600 s. Note that with the Eulerian scheme a time step of about 120 s would have been needed.



### 2.1.3 Boundary forcing

At the lateral boundaries, HIRHAM5 is currently driven by data from the ERA-Interim (ERAI) reanalysis (Dee et al., 2011). The ERAI data at 6-hourly resolution were downloaded from the Meteorological Archival and Retrieval System (MARS) at the European Centre for Medium-Range Weather Forecasts (ECMWF) and horizontally and vertically interpolated to the

regional model grid. The lateral boundary forcing includes all seven prognostic variables. All prognostic variables are treated equally using a 10-grid-point lateral boundary relaxation following Davies (1976). An inflow/outflow formulation for specific humidity, cloud water, and cloud ice is optionally available, but leads occasionally to numerical instabilities due to sharp differences between the imposed boundary values and the modeled values at the adjacent grid points.

At the lower boundary, HIRHAM5 uses sea surface temperature and sea-ice concentration from daily ERAI fields for all sea

grid points when running in stand-alone mode, or for the sea grid points outside the coupling domain when running in coupled mode with NAOSIM. Note that in coupled mode only a few HIRHAM5 sea grid points close to Pacific coasts are not covered by NAOSIM. Inside the coupling domain, sea surface temperature, sea-ice concentration, sea-ice thickness, snow thickness on ice, and the freezing temperature of sea water are transferred from NAOSIM to HIRHAM5 at 1-hour intervals. In stand-alone mode and outside the coupling domain, fixed values are used for the sea-ice thickness (2 m), the snow thickness on ice (0 m),

and the freezing temperature of sea water (271.38 K).

### 2.2 Ocean-sea ice model NAOSIM

The ocean–ice component of HN2.0 is the fine-resolution model (FRM), introduced by Fieg et al. (2010), from the North Atlantic/Arctic Ocean Sea-Ice Model (NAOSIM) hierarchy, whereas the earlier versions (HN1.x) have used the high-resolution model (HRM) of NAOSIM. A basic description of NAOSIM is given by Köberle and Gerdes (2003). Differences between FRM

and HRM are described by Fieg et al. (2010).

### 2.2.1 Ocean component

NAOSIM's ocean component is based on the Geophysical Fluid Dynamics Laboratory Modular Ocean Model MOM-2 (Pacanowski, 1996) and includes prognostic equations for horizontal velocity components, potential temperature, and salinity. The equations of motion are formulated in compliance with the Boussinesq, hydrostatic, and rigid lid approximations. The

horizontal velocity components are divided into a depth dependent internal mode velocity, representing the baroclinic flow, and a depth independent or external mode velocity, representing the barotropic flow. The rigid lid approximation at the ocean surface implies that the barotropic mode is non-divergent which allows the external mode velocity to be expressed in terms of a stream function. This barotropic stream function is obtained by iteratively solving the external mode elliptic equation using a conjugate gradient solver (see Pacanowski, 1996, for details).





### 2.2.2 Sea-ice component

The sea-ice component is based on the dynamic–thermodynamic sea ice model described by Harder et al. (1998) and represents an upgrade of the original Hibler (1979) model. Ice concentration, mean ice thickness (ice volume per area), mean snow thickness (snow volume per area), and ice age are computed from extended continuity equations and the ice drift velocities from the momentum balance where the inertia term is neglected. The internal stress is described by a viscous-plastic rheology according to Hibler (1979).

Thermodynamic processes are handled using the zero-layer approach by Semtner (1976). This means that there is no explicit snow or ice layer and neither snow nor ice temperatures are calculated. HN2.0 includes two thermodynamic ice growth schemes: The original scheme, where thermodynamic changes of ice and snow are derived from the energy balance of the combined ocean mixed layer–sea ice system (see Dorn et al., 2007), and an upgraded version of the flux-separating scheme of HN1.2 developed by Dorn et al. (2009). The main purpose of upgrading the flux-separating scheme was to allow for sublimation of snow and ice in the respective mass balances. This was neglected in HN1.2 due to the unavailability of corresponding coupling variables (see section 2.3). Besides, a few minor numerical improvements were implemented. The original scheme is currently the default for NAOSIM stand-alone simulations, while the flux-separating scheme is the default for coupled HIRHAM–NAOSIM simulations.

### 2.2.3 Ocean-sea ice coupling

The momentum coupling according to Hibler and Bryan (1987) involves a vertical integration of the momentum balance across both media, the sea ice and the uppermost ocean layer. Interfacial stresses do not appear explicitly and the prognostic variable is the mass-weighted mean velocity of both layers. Internal forces in the sea-ice component enter the overall momentum balance as a stress, much like the wind stress, the common driver of the melange. It is necessary to calculate one velocity separately to include new parameterizations of ocean–ice drag. Castellani et al. (2014) introduced a scheme that takes bottom and surface roughness of deformed ice into account (coded but not activated in HN2.0). In the current model configuration, sea-ice roughness is calculated with the Steiner et al. (1999) algorithm.

The benefit of the Hibler and Bryan (1987) approach is the representation of the total mass divergence near the surface and thus of the vertical Ekman velocities that are essential drivers of the large-scale ocean flow distant from lateral boundaries. On the other hand, the correct advection of oceanic constituents, including nutrients and other tracers, requires the separate knowlege of the ocean velocity. Especially in shallow waters, where the ice thickness becomes comparable to the water depth, uncertainties may arise.

### 2.2.4 Domain and discretization

NAOSIM's model domain encloses the entire Arctic Ocean, the Nordic seas, and the northern North Atlantic with the southern model boundary at approximately 50° N (see Figure 1). The model uses a rotated spherical coordinate system with the North Pole on the geographical equator at 60° E. The horizontal resolution is 1/12° (approximately 9 km). The equations are hori-



zontally discretized on an Arakawa B-grid, meaning that the horizontal velocity components are staggered compared to the tracer variables (i.e. temperature, salinity) by half a grid point distance in both longitude and latitude directions. For the tracers (velocities), the longitude range in the rotated pole grid is from 20.5417° W (20.5° W) to 39.7917° E (39.8333° E), and the latitude range is from 20.5417° S (20.5° S) to 21.5417° N (21.5833° N), resulting in two horizontal grids with 725×506 grid
points each. The grids are referred to as the T-grid (for the tracers) and the U-grid (for the velocities).

In the vertical, the model has 50 unevenly spaced $z$-coordinate levels. The depth of the levels and the corresponding vertical resolution is given in Table A2. Unlike in the horizontal, tracers and velocities are not staggered vertically and consequently discretized at the same depth.

The time stepping scheme of NAOSIM is a mixing scheme that mainly consists of a leapfrog time step. At regular intervals,
the leapfrog scheme is replaced by an Euler backward time step which is comprised of two half steps, a first forward step and a second step with data from the first step (also known as forward-backward method). Since the execution of two half steps is numerically more expensive, the interval of replacing the leapfrog by an Euler backward time step should be chosen as large as possible. Pacanowski (1996) indicates that a number of 17 time steps between mixing time steps (i.e. every 17th time step is an Euler backward time step, while the others are normal leapfrog time steps) has been empirically established.
This number mostly works well in NAOSIM, but the model tends to crash every now and then when solving the external mode elliptic equation for the barotropic stream function. It was found that a smaller number is able to avoid a model crash. Owing to these occasionally occurring model crashes, the here presented simulations were running with different number of mixing time steps, basically meaning that the simulation of the year in which the model crash occurred was repeated with a smaller number of mixing time steps. However, the time step of the simulations was always the same with 360 s.

**2.2.5   Boundary forcing**

At the southern model boundary in the northern North Atlantic, NAOSIM applies an open-boundary condition following the formulation of Stevens (1991). At inflow points, temperature and salinity are restored to the Levitus climatology (Levitus and Boyer, 1994; Levitus et al., 1994) with a time constant of 30 days. At outflow points, advection of tracers and sea ice is allowed as well as radiation of waves (Fieg et al., 2010). Other boundaries with substantial inflow of freshwater (Arctic rivers, Baltic
Sea, Hudson Bay, Bering Strait, and White Sea) are treated as virtual sinks of salt in the upper 30 m of the ocean (represented by the uppermost three levels).

At the upper boundary, NAOSIM uses daily means of 2-m air and dew-point temperature, cloud cover, precipitation, wind speed, and wind stress components from ERAI for all grid points when running in stand-alone mode, or for the grid points outside the coupling domain when running in coupled mode with HIRHAM5. The ERAI wind stress components were rotated
and interpolated to the regional U-grid, while all other ERAI variables were interpolated to the regional T-grid. Prior to the interpolation, ERAI data that represent land areas had been replaced by distance-weighted averages of data from the sea grid points. This was done to avoid the inclusion of data from land areas into the interpolation, because surface values over land may differ widely from those over sea, particularly in the case of existing coastal mountains.



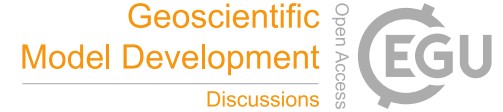

NAOSIM internally calculates the atmospheric fluxes from the prescribed atmospheric boundary values using standard bulk formulas. In addition, seven ice thickness classes, assuming a linear distribution between zero and twice the mean ice thickness, are used for calculating the atmospheric heat fluxes. This is done for all ocean grid points when running in stand-alone mode or only for the ocean grid points outside the coupling domain when running in coupled mode with HIRHAM5. Note that the

coupling domain does not cover a large part of the northern North Atlantic whereby the ERAI forcing is also relevant in coupled mode. Inside the coupling domain, the atmospheric fluxes are calculated by HIRHAM5 and transferred at 1-hour intervals (see also section 2.3). In HIRHAM5, all fluxes are calculated separately for the open water part and the ice-covered part of the grid cells, but without using ice thickness classes in the heat flux calculation, since this feature is not yet available in HIRHAM5 (see also section 4).

### 2.2.6   Technical modifications

As part of coupling NAOSIM's FRM version with HIRHAM5, NAOSIM was re-parallelized to enable its efficient operation on modern high-performance computer systems. Especially the inter-process communication could be reduced substantially. As a result, the runtime of NAOSIM simulations has been reduced by a factor of almost 10. In addition, restart issues were fixed with the side benefit that runs with intermediate restart produce now bit-identical results to continuous runs. This also

holds true now for coupled HIRHAM–NAOSIM simulations.

## 2.3   Coupling procedure

The coupling of HIRHAM5 and NAOSIM is handled by means of the relatively new and flexible coupling software YAC 1.2.0 (Hanke et al., 2016). YAC allows for parallelized interpolation and communication of coupling fields. The current interpolation stack for all coupling fields consists of a first-order conservative remapping (Jones, 1999) with a fractional area normalization

for the interpolation of partially covered source cells, followed by a distance-weighted two nearest neighbor interpolation. The downstream nearest neighbor interpolation enables allocating of values to all remaining non-land grid cells in case of non-overlapping cells due to the different resolutions and land-sea masks of HIRHAM5 and NAOSIM.

The time step of coupling (currently 1 hour for all fields) as well as optional time averaging of coupling fields is managed by YAC. In the current configuration, all fields that are transferred from HIRHAM5 to NAOSIM are averaged over the coupling

interval using YAC's time averaging option, while YAC transfers instantaneous fields from NAOSIM to HIRHAM5. The coupling fields are listed in Table 1. The fields that apply to the same source and target grids are transferred in bundles using the same interpolation weights. In HN2.0, the coupling fields are collected in 3 bundles: Bundle no. 1 comprises the fields to be transferred from the HIRHAM5-grid to NAOSIM's U-grid, bundle no. 2 comprises the fields to be transferred from the HIRHAM5-grid to NAOSIM's T-grid, and bundle no. 3 comprises the fields to be transferred from NAOSIM's T-grid to the

HIRHAM5-grid.

Wind stress and heat fluxes are provided by HIRHAM5 as potential fluxes (unweighted), while freshwater fluxes are provided as effective fluxes (weighted with the water/ice fraction). The latter is done in order to reduce the number of coupling fields by two. The freshwater flux over ocean thus includes large-scale plus convective rainfall (assuming that rainfall over ice is directly





**Table 1.** List of the coupling fields between HIRHAM5 and NAOSIM.

| Name | Short description | Source[a] | Bundle |
|---|---|---|---|
| taux_oce | U-wind stress ocean[b] | HIRHAM5 | 1 |
| tauy_oce | V-wind stress ocean[b] | HIRHAM5 | 1 |
| taux_ice | U-wind stress ice[b] | HIRHAM5 | 1 |
| tauy_ice | V-wind stress ice[b] | HIRHAM5 | 1 |
| fwat_oce | Freshwater flux ocean | HIRHAM5 | 2 |
| fwat_ice | Freshwater flux ice | HIRHAM5 | 2 |
| heat_oce | Heat flux ocean | HIRHAM5 | 2 |
| heat_ice | Heat flux ice | HIRHAM5 | 2 |
| sst | Sea surface temperature | NAOSIM | 3 |
| tfrezs | Freezing temperature | NAOSIM | 3 |
| a_ice | Sea-ice concentration | NAOSIM | 3 |
| h_ice | Sea-ice thickness[c] | NAOSIM | 3 |
| h_snow | Snow thickness on ice[c] | NAOSIM | 3 |

[a]Source indicates the model component in which the field is calculated; the recipient is always the other model component.

[b]Wind stress vectors are rotated prior to the transfer.

[c]Sea-ice thickness and snow thickness are transferred as actual thicknesses unlike the prognostic mean thicknesses (volume per area).

run off into the ocean) and the open water fraction of evaporation and large-scale plus convective snowfall. Snow falling in open water is directly melted. The required heat of fusion is subtracted from the heat flux over ocean. The freshwater flux over ice is consequently the ice fraction of sublimation and large-scale plus convective snowfall.

The coupling routines have been integrated in HIRHAM5 and NAOSIM via separate modules that provide an interface
between the respective model component and YAC (see Figure 2). These modules comprise all initialization and working routines required for the exchange of data between the two model components using YAC. To avoid pole problems inherent with the use of conventional geographic coordinates, NAOSIM's rotated pole grid is treated for the coupling as being a regular (but limited) latitude-longitude grid and HIRHAM's longitudes and latitudes are translated into these grid coordinates. The fact that the two rotated grids have their intersection point with the cartesian $z$-axis at coordinates $(0,0)$, the geographic North
Pole, the coordinate transformation as well as the wind stress rotation can be achieved by a single rotation around this $z$-axis (corresponds to the cartesian $x$-axis when treating the NAOSIM grid as regular grid). In this way, the grid transformation of the coupling fields remains numerically as efficient as possible.

The coupling domain is basically defined as the overlap area of the components' model domains (see Figure 1). Grid cells that represent land are excluded and masked as uncoupled cells. The coupling domain in NAOSIM is limited by the outer vertices
of the outermost HIRHAM5 grid cells. In HIRHAM5 coordinates, these vertices are at longitudes 27.50° W and 27.00° E and





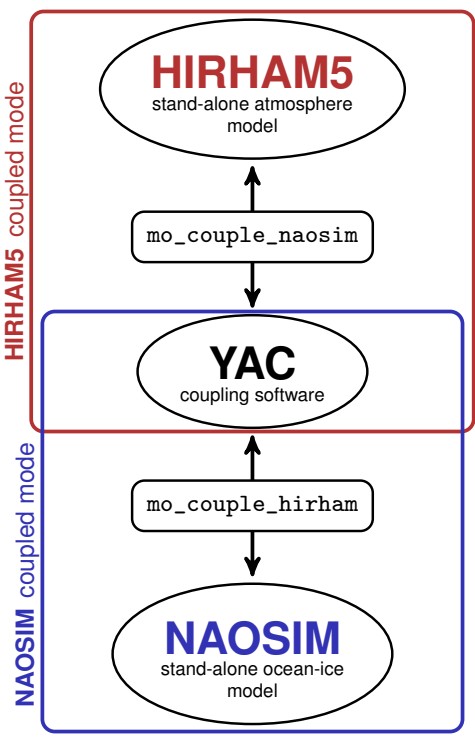

**Figure 2.** Schematic diagram of the coupling of HIRHAM5 and NAOSIM using the coupling software YAC. The boxes "mo_couple_naosim" and "mo_couple_hirham" represent the interface modules between the two model components and YAC.

at latitudes $25.25°$ S and $24.75°$ N. NAOSIM grid cells whose HIRHAM5-transformed longitude and latitude fall outside this area are masked as uncoupled cells.

For the coupling domain in HIRHAM5, the outermost NAOSIM grid cells are neglected, since these cells mostly represent a passive boundary. Since HIRHAM5 only receives fields from NAOSIM's T-grid, the coupling domain in HIRHAM5 is thus

5   limited by the outer vertices of the second outermost NAOSIM T-grid cells. In NAOSIM coordinates, these vertices are at longitudes $20.50°$ W and $39.75°$ E and at latitudes $20.50°$ S and $21.50°$ N. Like in NAOSIM, HIRHAM5 grid cells whose NAOSIM-transformed longitude and latitude fall outside this area are masked as uncoupled cells. In both model components, masked grid cells are treated as in stand-alone mode.

## 3   Evaluation of the base configuration

10   HIRHAM5 as well as NAOSIM can be configured via a number of namelist parameters. For the first coupled simulations with HN2.0, these namelist parameters were carried over from the stand-alone model versions except for the modifications explicitly specified in section 2. Another exception is that the coupled mode flag is activated in the ECHAM5 part of the model. This has the side effect that different default parameters are used in the ECHAM5 physics. In particular, the mixing ratios of greenhouse





gases are preset in coupled mode with values for the year 1860. This was actually not intended and basically means that the local greenhouse effect is underestimated in the current HN2.0 simulations which should actually represent the present-day climate. The impact of time-adjusted greenhouse gases will be evaluated at a later date in a new configuration. The current setup of HN2.0 is here taken as a basis and simply referred to as the base configuration. Relevant physical constants used in

the base configuration are given in Table B1.

## 3.1    Ensemble simulation setup

An ensemble of 10 hindcast simulations were carried out with HN2.0 for the period 1979–2016. The simulations were driven by ERAI data as detailed in section 2. All ensemble members were equally started on 2 January 1979 and run through 31 December 2016. All atmospheric fields were initialized with the corresponding ERAI fields using implicit normal mode initialization

(Temperton, 1988), while the initial ocean and sea-ice fields were taken from the Januaries 1991 to 2000 of a preceding coupled spin-up run for the period 1979–2000. The coupled spin-up run itself were initialized with ocean and sea-ice fields of a 20-year-long NAOSIM run which started from rest. The coupled spin-up run as well as the uncoupled NAOSIM run were driven by ERAI too.

## 3.2    Data for model evaluation

Simulated sea ice is compared against sea-ice thickness data from the Pan-Arctic Ice-Ocean Modeling and Assimilation System (PIOMAS) described by Schweiger et al. (2011) and against sea-ice concentrations from Nimbus-7 SMMR and DMSP SSM/I-SSMIS passive microwave data (Cavalieri et al., 1996, updated 2017). PIOMAS data are available via ftp://pscftp.apl.washington.edu/zhang/PIOMAS/data/v2.1/ and the satellite-derived sea-ice concentrations (hereinafter simply referred to as satellite data) from the web-site http://nsidc.org/data/NSIDC-0051.

Atmospheric temperatures are compared against the ERAI data, also used as boundary forcing as described in section 2. The ERAI temperatures are relatively well constrained by the data assimilation system (Dee et al., 2011) and can be considered as quasi realistic, even though there is a warm wintertime bias in surface air temperature over Arctic sea ice (Simmons and Poli, 2015). Nevertheless, the near-surface air temperature bias compared to observations is in ERAI smaller than in other reanalysis products (Lindsay et al., 2014).

In addition, the HN2.0 ensemble simulations are compared against a 10-member ensemble of ERAI-driven simulations with HN1.2. This ensemble covers the period from 1979 to 2014 and has been described by Graham et al. (2017) and Rinke et al. (2018). Even though the HN1.2 ensemble setup differs slightly from the current setup, the forcing data are identical. Differences in the simulation results between the ensembles of HN1.2 and HN2.0 thus indicate changes in the model performance.





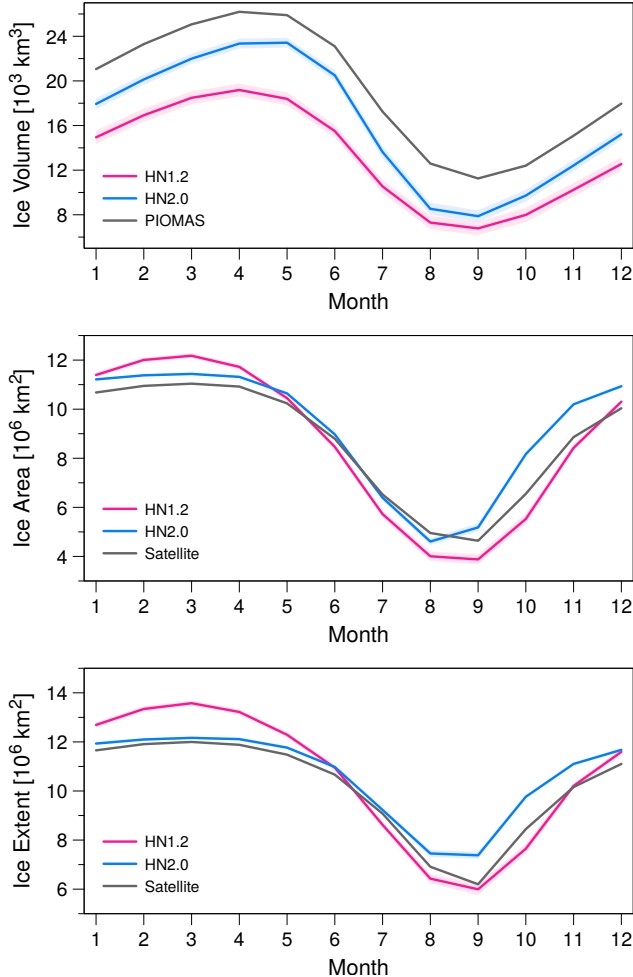

**Figure 3.** Mean seasonal cycle of sea-ice volume (top), sea-ice area (middle), and sea-ice extent (bottom). The red lines represent the ensemble mean of the 10-member ensemble of ERAI-driven simulations with HN1.2 and the very narrow red shaded areas the two-standard-deviation range of the ensemble. Analogously, the blue lines and blue shaded areas represent the HN2.0 ensemble. The gray lines represent sea-ice volume from PIOMAS and sea-ice area and extent from the SMMR and SSM/I-SSMIS satellite data. PIOMAS and satellite data were reduced to the NAOSIM domain. Data voids around the North Pole in the satellite data were filled with distance-weighted averages. HN1.2 data refer to the period 1979–2014, satellite data to the period 1979–2015, and all other data to the period 1979–2016.

## 3.3 Sea-ice climatology

Sea ice is an important indicator for the overall performance of a coupled Arctic climate model, because it depends on atmospheric and oceanic processes to a similarly large extent. The focus of the evaluation is therefore on the simulated mean sea-ice state.



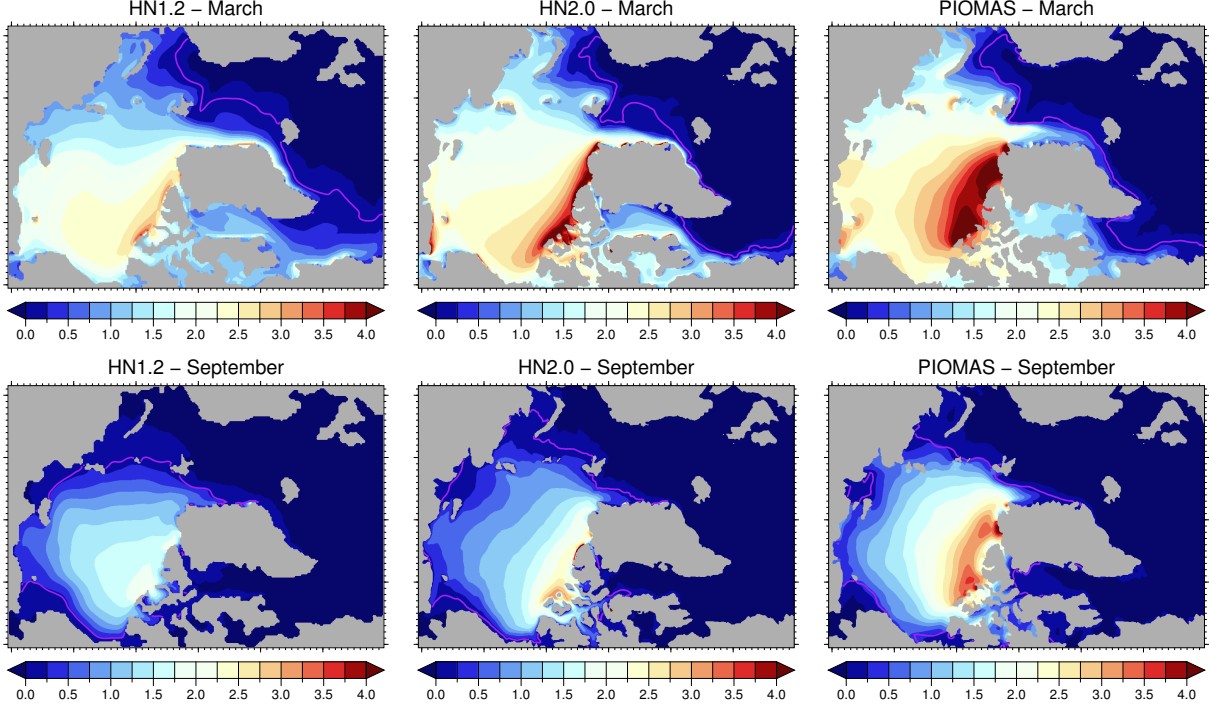

**Figure 4.** Sea-ice thickness climatology (in meters) for March (top) and September (bottom) from ERAI-driven ensemble simulations with HN1.2 (left; period 1979–2014), ERAI-driven ensemble simulations with HN2.0 (middle; period 1979–2016), and PIOMAS data (right; period 1979–2016). PIOMAS data were interpolated to the T-grid of HN2.0. The purple lines represent the climatological 15 % sea-ice concentration contour of the respective dataset. Land areas of the T-grid are shown in gray. Note that the T-grids of HN1.2 and HN2.0 slightly differ due to the different horizontal resolution.

The mean seasonal cycle of sea-ice volume, sea-ice area, and sea-ice extent is shown in Figure 3 (sea-ice extent is here defined as area of all grid cells with more than 15 % sea-ice concentration). The across-ensemble scatter in the mean seasonal cycle of all these variables is generally low. This indicates that the climatological differences between HN1.2 and HN2.0 are robust and statistically reliable. HN2.0 simulates an ice volume which is in all months much closer to the PIOMAS data than those from the HN1.2 simulations, even though the PIOMAS ice volume is still higher throughout the year. Largest differences between HN2.0 and PIOMAS appear in August and September as a result of around 1 m thinner ice in the central Arctic in HN2.0. However, despite thinner ice, the geographical distribution of relatively thick and thin ice in HN2.0 agrees rather well with PIOMAS, representing a clear improvement as compared to HN1.2 (see Figure 4).

With respect to ice area and ice extent, HN2.0 shows rather good agreement with the satellite data from Januar to August. Especially the considerable and permanent overestimate of sea ice in the Labrador Sea during winter in HN1.2 (Figure 4; top left) does not appear in HN2.0 any longer. Instead, sea ice in the Labrador Sea is even a little underestimated as compared to

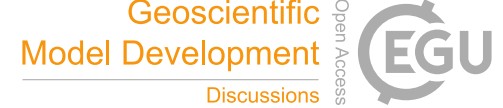



PIOMAS and satellite data. In contrast, the ice extent in the Greenland and Barents seas is still overestimated during the cold season, but this bias is reduced as compared to HN1.2.

A less good agreement between HN2.0 and the satellite data appears from September to December. The increase in ice area and ice extent appears too early and is stronger than in the satellite data. Larger parts of the Kara and Chukchi seas are
on average ice covered in September, and the overestimate of ice area and ice extent persists until December. One possible reason for this overestimate could be reduced values of the reference thicknesses for lateral freezing as compared to HN1.2 (see Table B1). The original values introduced by Dorn et al. (2009) had been resulted in generally too thick sea ice in HN2.0 and were reduced for that reason. Fine-tuning of these parameters is intended as part of generating a new configuration.

### 3.4  Sea-ice trends and variability

Temporal trends of sea-ice volume, sea-ice area, and sea-ice extent are shown in Figure 5. The downward trend in the PIOMAS and satellite data is only rudimentarily reproduced by HN1.2 and even still less by HN2.0. However, HN2.0 shows reasonable agreement with the PIOMAS ice volume at the end of the simulation period, and some agreement in ice area and extent with the satellite data in the first half of the simulation period. In contrast, HN1.2 shows better agreement in September ice area and extent at the end of the simulation period. Given that the ocean model uses climatological boundary forcing and that greenhouse gases are constant in the two ensembles, the underestimate of the observed downward trend in sea ice is not a
surprise. The existing, but weak downward trend in sea-ice volume, area, and extent in the ensemble simulations can thus only arise from large-scale atmospheric changes entering the model via the atmospheric model boundaries, in combination with internal atmosphere-ocean-ice feedbacks.

In order to detect coherent variability in the model data and the observational-like data, it is useful to evaluate the detrended
time series. On that account, correlation coefficients between the detrended time series of the PIOMAS and satellite data on the one hand and the model data on the other hand have been calculated. The results are listed in Table 2 for each month of the detrended ice volume time series. With respect to the ice volume, HN2.0 shows significantly higher correlations with the PIOMAS data than HN1.2. This applies to all months, to the mean correlation of the ensemble members, and to the correlation of the ensemble mean, and clearly indicates an improved year-to-year variability in HN2.0.

In comparison to the low ensemble scatter with respect to the mean seasonal cycle, the ensemble scatter for individual years is relatively large (see also Figure 5). The strongest ensemble scatter in ice area and extent appears in the summer months with deviations of up to $1.3 \cdot 10^6$ km$^2$, while the ensemble member deviations in ice volume can amount up to $2.5 \cdot 10^3$ km$^3$ throughout the year. Generally, the scatter in the HN2.0 ensemble is in most years a little larger than in HN1.2. The large ensemble scatter in individual years emphasizes the importance of internal variability in the model. Even though the model
is constrained at the lateral boundaries, it is able to generate different sea-ice states in its interior with the same forcing from the exterior, as demonstrated in Figure 6, showing the simulated Arctic sea-ice thickness distribution in September 2016 from two ensemble members. Therefore, the model is not necessarily able to reproduce observed sea-ice extrema when inner-Arctic processes contributed to that. For instance, the all-time sea-ice minimum in 2012, which was partly caused by a strong storm





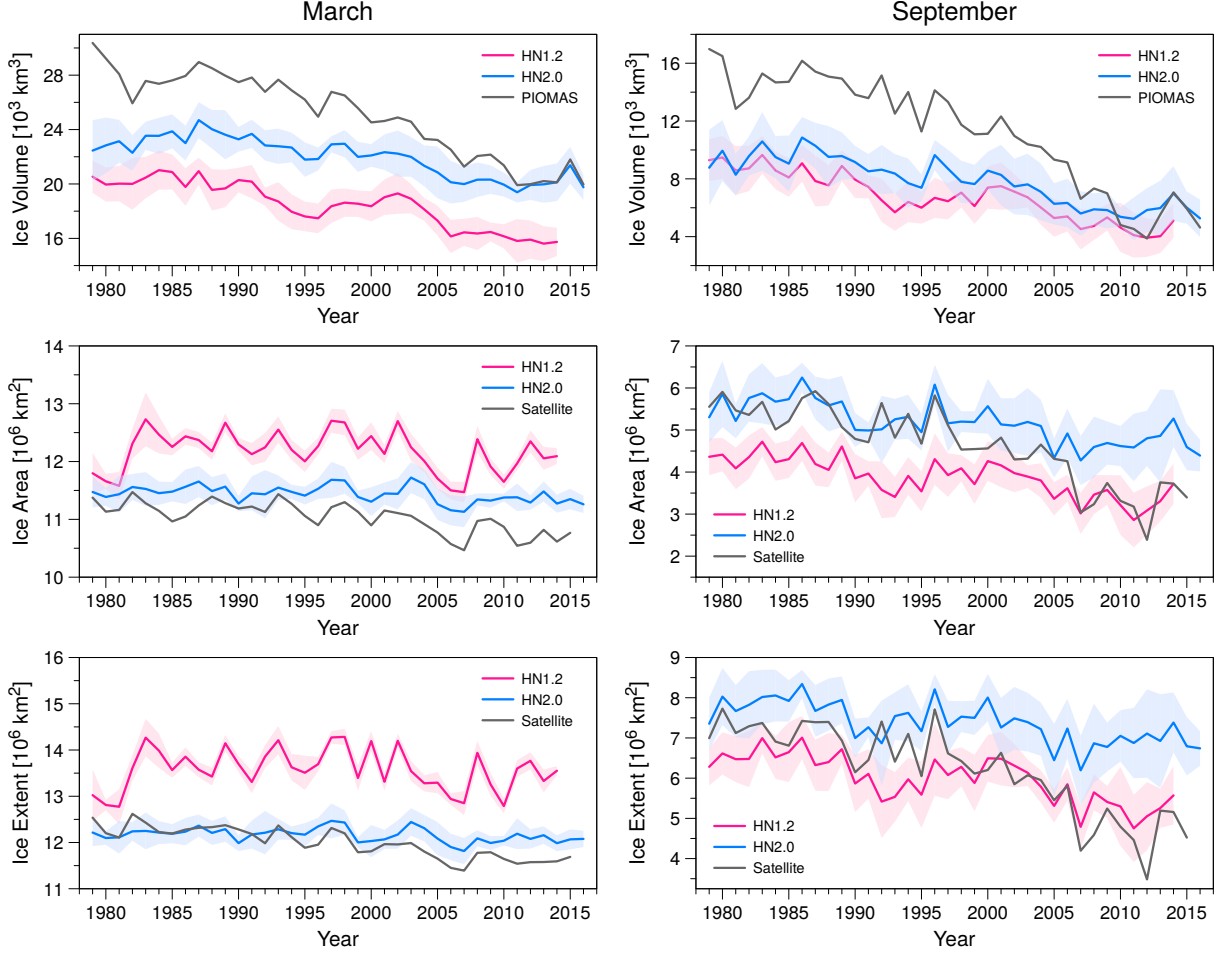

**Figure 5.** Temporal trends of sea-ice volume (top), sea-ice area (middle), and sea-ice extent (bottom) in March (left column) and September (right column) from 1979 to 2016. The red lines represent the ensemble mean of the 10-member ensemble of ERAI-driven simulations with HN1.2 and the red shaded areas the two-standard-deviation range of the ensemble. Analogously, the blue lines and blue shaded areas represent the HN2.0 ensemble. The gray lines represent sea-ice volume from PIOMAS and sea-ice area and extent from the SMMR and SSM/I-SSMIS satellite data. PIOMAS and satellite data were reduced to the NAOSIM domain. Data voids around the North Pole in the satellite data were filled with distance-weighted averages.

that entered the central Arctic in early August 2012 (e.g. Parkinson and Comiso, 2013), is not reproduced by the model, neither by HN1.2 nor by HN2.0.



**Table 2.** Correlation coefficients between the detrended monthly time series of the simulated sea-ice volume and the PIOMAS sea-ice volume. The term "Mean correlation" indicates the mean correlation coefficients between PIOMAS and each of the 10 individual ensemble members; the term "Ensemble mean" indicates the correlation coefficients between PIOMAS and the ensemble mean time series.

| Month | Mean correlation | | Ensemble mean | |
|---|---|---|---|---|
| | HN1.2 | HN2.0 | HN1.2 | HN2.0 |
| 1 | 0.26 | 0.56 | 0.31 | 0.67 |
| 2 | 0.33 | 0.60 | 0.39 | 0.72 |
| 3 | 0.33 | 0.60 | 0.39 | 0.71 |
| 4 | 0.33 | 0.61 | 0.38 | 0.72 |
| 5 | 0.30 | 0.58 | 0.35 | 0.68 |
| 6 | 0.36 | 0.52 | 0.42 | 0.62 |
| 7 | 0.40 | 0.52 | 0.48 | 0.65 |
| 8 | 0.29 | 0.52 | 0.35 | 0.66 |
| 9 | 0.28 | 0.52 | 0.34 | 0.66 |
| 10 | 0.30 | 0.55 | 0.36 | 0.69 |
| 11 | 0.29 | 0.58 | 0.36 | 0.73 |
| 12 | 0.27 | 0.61 | 0.32 | 0.76 |

## 3.5 Near-surface air temperatures

### 3.5.1 Winter

Wintertime near-surface air temperatures are naturally strongly affected by the surface type, particularly whether the surface is land, ocean, or sea ice. Consequently, the largest winter temperature biases in HN2.0 coincide with the largest biases in sea-ice
extent. Figure 7 (top) shows that the mean January temperatures are 7–8 K colder over the northern Barents Sea, associated with the overestimate of sea ice in this region, and 5–6 K warmer over the Labrador Sea, associated with the underestimate of sea ice there. More conspicuous is that HN2.0 shows 2–4 K colder January temperatures over the ice-covered Arctic Ocean compared to ERAI.

The winter temperature biases in HN1.2 are totally different. On the one hand, the January temperatures are 1–2 K colder over
the Labrador Sea, associated with the permanent overestimate of sea ice in this region, on the other hand, HN1.2 shows 4–8 K warmer January temperatures over the ice-covered Arctic Ocean as compared to ERAI. This means that there are differences between HN2.0 and HN1.2 of up to 12 K over sea ice, while temperature differences over land are relatively small (less than 5 K). Except for the Labrador Sea and the sea region around the southern tip of Greenland, winter temperatures are in HN2.0 colder than in HN1.2. Given that HN2.0 has erroneously used pre-industrial greenhouse gas concentrations, the colder
temperatures might be explained to some degree by the associated reduced longwave radiative forcing. However, the warm temperature bias over sea ice in HN1.2 is definitely a major shortcoming of the old model version that does not anymore



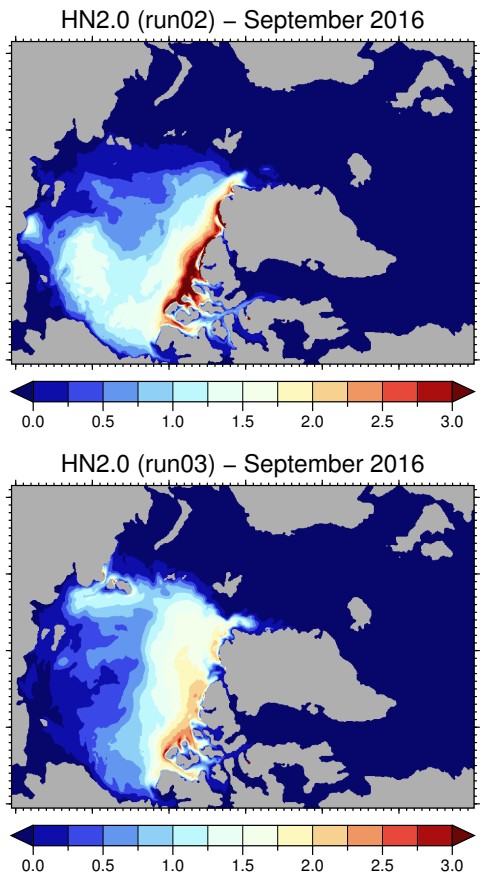

**Figure 6.** Mean sea-ice thickness (in meters) in September 2016 from two ensemble members, labeled as run02 (top) and run03 (bottom), of the ERAI-driven ensemble simulations with HN2.0, as an example for the coupled model's ability to generate different sea-ice states under identical boundary conditions. Land areas of the T-grid are shown in gray.

appear in the new model version. Also considering that ERAI has a warm wintertime bias in surface air temperature over Arctic sea ice (Simmons and Poli, 2015), a little colder temperatures in HN2.0 represent an improvement compared to the significant warmer temperatures in HN1.2.

### 3.5.2 Summer

5 Summertime near-surface air temperatures are usually close to the freezing temperature of water over Arctic sea ice. This holds for the July temperatures in ERAI which are on average a little above $0\,°C$ in the central Arctic (Figure 7; bottom). Both HN2.0 and HN1.2 show on average temperatures below $0\,°C$ in this region and are here around $1.3\,K$ colder than ERAI. The reason for that is unclear. It could possibly be associated with an approximately by $10\,\%$ underestimated sea-ice concentration (not shown) and thus more open water area with surface temperatures at the salinity-dependent freezing temperature of sea water
10 (usually between -1.8 and $-1.4\,°C$).



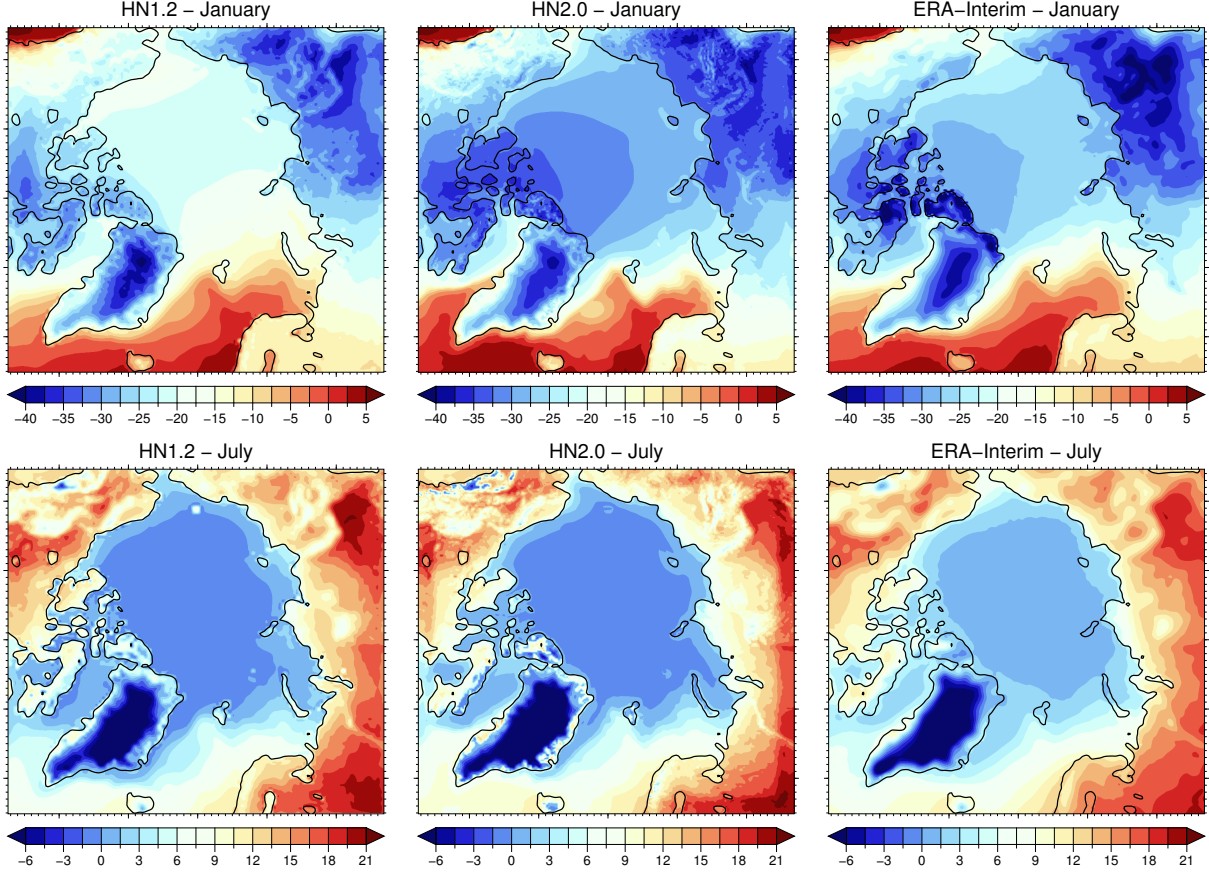

**Figure 7.** 2-m air temperature climatology (in degrees Celsius) for January (top) and July (bottom) from ERAI-driven ensemble simulations with HN1.2 (left), ERAI-driven ensemble simulations with HN2.0 (middle), and ERAI data (right). ERAI data were interpolated to the HIRHAM5-grid. The black lines represent the coastlines of HIRHAM5, defined as 50 % land-sea fraction. All data refer to the period 1979–2014.

Even though this temperature bias is low, it might have consequences for the way how sea ice is melting in the model. Due to the air temperatures below $0\,^\circ$C melting of the snow/ice pack from above is reduced and melt ponds are underrepresented. Instead, sea ice is rather melting from below through increased heat fluxes entering the upper ocean layer due to the larger open water area. Potentially, a restriction of the decrease in sea-ice concentration when melting conditions occur could contribute to

5   reduce both the open water bias and the temperature bias. This will be subject of forthcoming studies.

Another systematic bias of HN2.0 appears in the boundary zone, where the model overestimates the near-surface air temperatures. The reason for this warm temperature bias is a permanent underestimate of clouds in the boundary zone, in which specific humidity, cloud water, and cloud ice are relaxed to ERAI data. This bias is not a special feature of the coupled model, but appears in HIRHAM5 stand-alone simulations as well. Several efforts have been undertaken to overcome this problem, as

10   for instance different boundary relaxation functions, changes of the boundary zone width, an inflow/outflow formulation for

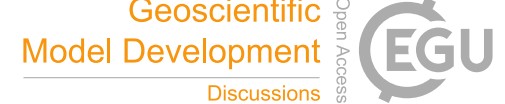



specific humidity, cloud water, and cloud ice at the boundary, modifications in the cloud parameterization, and others, but the boundary problem could not be solved until now. Only the inflow/outflow formulation helped to restrict the problem to the outermost boundary points.

## 4 Conclusions

The upgraded version of the coupled Arctic atmosphere-ocean-sea ice model HIRHAM–NAOSIM (HN2.0) has been described, and the current base configuration of HN2.0 has been evaluated against observationally based data sets. As compared to the previous version (HN1.2), the upgraded version shows a number of improvements, especially a generally thicker Arctic sea-ice cover where the sea-ice thickness distribution is in better agreement with the PIOMAS data set. Wintertime biases in sea-ice extent and near-surface air temperatures are reduced as well, while summertime biases are of similar magnitude as in the previous version.

However, the choice of parameters for the current base configuration of HN2.0 has turned out to be still in need of improvement. A couple of parameter adaptations and their tendential effect on the model results have already been tested in HN2.0; however, as already known from previous work (Dorn et al., 2009), an adequate harmonization of various parameters is needed to reduce model biases in the end. In particular, the mixing ratios of greenhouse gases, the reference thicknesses for lateral freezing of sea ice, and the fractions of snow and melt ponds need to be reassessed and harmonized with other parameters that are not well established by observations. Such a reassessment and harmonization of model parameters is currently in progress and will eventually lead to a revised model configuration that might show an even better performance.

Moreover, the upgraded version of HIRHAM–NAOSIM has been designed as a tool for advancing our understanding of interaction processes between atmosphere, sea ice, and ocean in combination with observations. Simple fine-tuning of individual parameters/parameterizations towards better model performance is therefore not the primary objective. Instead, more realistic parameterizations should be developed on the basis of observations.

A short-term objective is the implementation of the stability-dependent parameterization of the transfer coeffcients for momentum and heat over sea ice by Lüpkes and Gryanik (2015), which also accounts for to date disregarded form drag caused by the edges of ice floes and melt ponds. The translation of this parameterization into model source code has recently been completed. Sensitivity studies with respect to this new parameterization will be carried out in the near future. It is anticipated that the additional form drag of the sea ice will lead to improvements in the sea-ice drift.

The inclusion of ice thickness classes in the heat flux calculation of the atmospheric component might be a useful step to improve the thermodynamic coupling between atmosphere and sea ice. Castro-Morales et al. (2014) showed that even a change from seven uniform ice thickness classes (as currently implemented in NAOSIM) to fifteen nonuniform ice thickness classes (derived from field measurements) has the potential to improve the representation of the sea-ice energy balance considerably. Conceptions how such a nonuniform ice thickness distribution can be implemented in the atmospheric component HIRHAM5 are under discussion.



Furthermore, first attempts to derive an improved parameterizations of the fractions of snow and melt ponds on sea ice from data of the combined ACLOUD/PASCAL field campaigns (Wendisch et al., 2018) are underway. It already became apparent during the setup of the base configuration that the ratio of these fractions determines the surface albedo in early summer with significant impact on the total extent of thermodynamic sea-ice loss during the summer. An improved, observationally based parameterization of the sea-ice surface fractions is therefore of prime importance.

The ACLOUD/PASCAL data are also useful to reconsider the cloud parameterization, particularly in terms of the microphysical properties and the cloud coverage. Although the cloud feedback is not rated as the largest contributor to Arctic amplification (Taylor et al., 2013; Pithan and Mauritsen, 2014), it belongs to the largest uncertainties in coupled climate models (Soden and Held, 2006). Improvements in the simulation of Arctic clouds due to the modifications suggested by Klaus et al. (2016) were already achieved, but further improvements on the basis of the ACLOUD/PASCAL data would be desirable, given the important role of clouds in the radiation budget. This will be subject of future developments.

*Code availability.* The model source code is available from the AWIForge repository (https://swrepo1.awi.de/projects/hirham-naosim/). Access to the repository will be granted to individuals on request. Registered individuals can access the code using the open source subversion software (http://subversion.apache.org/). HN2.0, as described here, corresponds to revision 242 in the repository. Because HN2.0 contains source code being subject to intellectual property rights, any user of the HN2.0 source code need to comply with the conditions of use for HIRLAM source code (see http://hirlam.org/index.php/hirlam-programme-53/access-to-the-models) and for ECHAM source code (see http://www.mpimet.mpg.de/en/science/models/license/).

## Appendix A: Vertical levels in HIRHAM5 and NAOSIM as used in HN2.0

HIRHAM's hybrid terrain-following/pressure levels are defined by the pressures of the interfaces between them (the so-called half levels). These pressures are given by

$$p_{k+1/2} = A_{k+1/2} + B_{k+1/2} \cdot p_s \tag{A1}$$

where $p_s$ is the surface pressure. $A_{k+1/2}$ and $B_{k+1/2}$ are constants whose values ultimately determine the vertical level with index $k$. The central pressures of the levels (at the so-called full levels) can be simply obtained by arithmetic averaging of the interface pressures:

$$p_k = \frac{1}{2}(p_{k+1/2} + p_{k-1/2}). \tag{A2}$$

The constants $A_{k+1/2}$ and $B_{k+1/2}$ of HIRHAM's 40-layer version, which is used in the current configuration of HN2.0, are listed in Table A1. The corresponding half- and full-level pressures were computed for a surface pressure $p_s = 101325\,\mathrm{Pa}$ from (A1) and (A2). The full-level-related geometric heights $z_k$ were here roughly estimated by means of the simplified barometric formula

$$z_k = -H \ln\left(\frac{p_k}{p_s}\right) \tag{A3}$$





with the scale height $H = 8000\ \mathrm{m}$. Note that this simplification is not used in HIRHAM5.

NAOSIM's vertical levels are defined as geometric depths below the sea surface ($z$-coordinate levels). The depth of the 50 levels and the corresponding vertical resolution used in HN2.0 is given in Table A2.

**Appendix B:  The base configuration of HN2.0**

The base configuration of HN2.0 is defined by a specific choice of parameters. On the one hand, these are physical constants as part of the physical parameterizations, on the other hand, it is the choice of the parameterizations itself, in case alternative parameterization packages are available. Since there is a vast number of physical constants in the two model components, it is rather impossible to list them all. Instead, we concentrate here on the most relevant physical constants that are of particular importance for the interaction processes between the two model components. These constants are listed in Table B1. In order

to ease future changes of the constants, the respective subroutines or namelists, where the constants are specified, are indicated in addition to the current values. A couple of constants are relevant to both model components, even if not always explicitly indicated.

Alternative parameterizations are activated in HN2.0 either by namelist parameters or by preprocessor definitions. A list of the namelist parameters of the ECHAM5 physics (see Roeckner et al., 2003, for a detailed description of the physical

parameterizations) is given in Table B2, and the preprocessor definitions used to build the two components of HN2.0 are listed in Table B3. The base configuration of HN2.0 uses mostly the default settings, unless otherwise specified in section 2.





**Table A1.** Vertical-coordinate parameters ($A_{k+1/2}$; $B_{k+1/2}$) of HIRHAM's 40-layer version as used in HN2.0. The associated half- and full-level pressures ($p_{k+1/2}$; $p_k$) and geometric heights ($z_k$) apply to a surface pressure $p_s = 101325$ Pa and a scale height $H = 8000$ m.

| $k$ | $A_{k+1/2}$ [Pa] | $B_{k+1/2}$ | $p_{k+1/2}$ [Pa] | $p_k$ [Pa] | $z_k$ [m] |
|---|---|---|---|---|---|
| 0 | 0.000000 | 0.0000000000 | 0 | – | – |
| 1 | 2000.000000 | 0.0000000000 | 2000 | 1000 | 36947 |
| 2 | 4000.000000 | 0.0000000000 | 4000 | 3000 | 28158 |
| 3 | 6000.000000 | 0.0000000000 | 6000 | 5000 | 24071 |
| 4 | 8000.000000 | 0.0000000000 | 8000 | 7000 | 21379 |
| 5 | 9988.883000 | 0.0001971156 | 10009 | 9004 | 19365 |
| 6 | 11914.520000 | 0.0015112920 | 12068 | 11038 | 17736 |
| 7 | 13722.940000 | 0.0048841570 | 14218 | 13143 | 16340 |
| 8 | 15369.730000 | 0.0110761700 | 16492 | 15355 | 15095 |
| 9 | 16819.480000 | 0.0206778900 | 18915 | 17703 | 13957 |
| 10 | 18045.180000 | 0.0341211600 | 21503 | 20209 | 12898 |
| 11 | 19027.700000 | 0.0516904100 | 24265 | 22884 | 11903 |
| 12 | 19755.110000 | 0.0735338300 | 27206 | 25736 | 10964 |
| 13 | 20222.210000 | 0.0996747000 | 30322 | 28764 | 10074 |
| 14 | 20429.860000 | 0.1300225000 | 33604 | 31963 | 9230 |
| 15 | 20384.480000 | 0.1643843000 | 37041 | 35323 | 8430 |
| 16 | 20097.400000 | 0.2024760000 | 40613 | 38827 | 7674 |
| 17 | 19584.330000 | 0.2439331000 | 44301 | 42457 | 6959 |
| 18 | 18864.750000 | 0.2883229000 | 48079 | 46190 | 6285 |
| 19 | 17961.360000 | 0.3351549000 | 51921 | 50000 | 5650 |
| 20 | 16899.470000 | 0.3838922000 | 55797 | 53859 | 5056 |
| 21 | 15706.450000 | 0.4339629000 | 59678 | 57738 | 4499 |
| 22 | 14411.120000 | 0.4847716000 | 63531 | 61604 | 3981 |
| 23 | 13043.220000 | 0.5357099000 | 67324 | 65427 | 3499 |
| 24 | 11632.760000 | 0.5861684000 | 71026 | 69175 | 3054 |
| 25 | 10209.500000 | 0.6355475000 | 74606 | 72816 | 2643 |
| 26 | 8802.355000 | 0.6832686000 | 78035 | 76320 | 2267 |
| 27 | 7438.805000 | 0.7287858000 | 81283 | 79659 | 1925 |
| 28 | 6144.316000 | 0.7715966000 | 84326 | 82805 | 1615 |
| 29 | 4941.777000 | 0.8112534000 | 87142 | 85734 | 1337 |
| 30 | 3850.913000 | 0.8473749000 | 89711 | 88427 | 1089 |
| 31 | 2887.697000 | 0.8796569000 | 92019 | 90865 | 872 |
| 32 | 2063.780000 | 0.9078839000 | 94055 | 93037 | 683 |
| 33 | 1385.913000 | 0.9319403000 | 95815 | 94935 | 521 |
| 34 | 855.361800 | 0.9518215000 | 97299 | 96557 | 386 |
| 35 | 467.333500 | 0.9676452000 | 98514 | 97906 | 275 |
| 36 | 210.393900 | 0.9796627000 | 99475 | 98994 | 186 |
| 37 | 65.889240 | 0.9882701000 | 100202 | 99839 | 118 |
| 38 | 7.367743 | 0.9940194000 | 100726 | 100464 | 68 |
| 39 | 0.000000 | 0.9976301000 | 101085 | 100906 | 33 |
| 40 | 0.000000 | 1.0000000000 | 101325 | 101205 | 9 |




**Table A2.** Depth ($z$) and representing layer thickness ($\Delta z$) of NAOSIM's vertical levels as used in HN2.0.

| Level | $z$ [m] | $\Delta z$ [m] | Level | $z$ [m] | $\Delta z$ [m] |
|---|---|---|---|---|---|
| 1 | 5.00 | 10.00 | 26 | 271.58 | 23.59 |
| 2 | 15.00 | 10.00 | 27 | 299.52 | 32.28 |
| 3 | 25.00 | 10.00 | 28 | 337.12 | 42.94 |
| 4 | 35.00 | 10.00 | 29 | 386.30 | 55.42 |
| 5 | 45.00 | 10.00 | 30 | 448.81 | 69.58 |
| 6 | 55.00 | 10.00 | 31 | 526.21 | 85.24 |
| 7 | 65.00 | 10.00 | 32 | 619.93 | 102.19 |
| 8 | 75.00 | 10.00 | 33 | 731.14 | 120.23 |
| 9 | 85.00 | 10.00 | 34 | 860.83 | 139.14 |
| 10 | 95.00 | 10.00 | 35 | 1009.73 | 158.66 |
| 11 | 105.00 | 10.00 | 36 | 1178.34 | 178.56 |
| 12 | 115.00 | 10.00 | 37 | 1366.91 | 198.58 |
| 13 | 125.00 | 10.00 | 38 | 1575.44 | 218.48 |
| 14 | 135.00 | 10.00 | 39 | 1803.69 | 238.01 |
| 15 | 145.00 | 10.00 | 40 | 2051.14 | 256.91 |
| 16 | 155.00 | 10.00 | 41 | 2317.07 | 274.95 |
| 17 | 165.00 | 10.00 | 42 | 2600.50 | 291.91 |
| 18 | 175.00 | 10.00 | 43 | 2900.23 | 307.56 |
| 19 | 185.00 | 10.00 | 44 | 3214.87 | 321.72 |
| 20 | 195.00 | 10.00 | 45 | 3542.84 | 334.21 |
| 21 | 205.00 | 10.00 | 46 | 3882.37 | 344.86 |
| 22 | 215.00 | 10.00 | 47 | 4231.58 | 353.55 |
| 23 | 225.14 | 10.28 | 48 | 4588.44 | 360.16 |
| 24 | 236.54 | 12.52 | 49 | 4950.83 | 364.62 |
| 25 | 251.29 | 16.98 | 50 | 5316.57 | 366.86 |



**Table B1.** Physical constants used in the base configuration of HN2.0.

| Constant | Value | [Unit] | Specified by (subroutine(s)/*namelist*)[a] |
|---|---|---|---|
| Earth radius | 6371000 | m | hard-coded (mo_constants) |
| Gravity acceleration | 9.80665 | $\mathrm{m\,s^{-2}}$ | hard-coded (mo_constants) |
| Solar constant | 1367 | $\mathrm{W\,m^{-2}}$ | hard-coded (setrad) |
| $CO_2$ volume mixing ratio | 286.2 | ppmv | default (setrad/*radctl*) |
| $CH_4$ volume mixing ratio | 805.6 | ppbv | default (setrad/*radctl*) |
| $N_2O$ volume mixing ratio | 276.7 | ppbv | default (setrad/*radctl*) |
| Gas constant for dry air | 287.05 | $\mathrm{J\,kg^{-1}K^{-1}}$ | hard-coded (mo_constants) |
| Gas constant for water vapour | 461.51 | $\mathrm{J\,kg^{-1}K^{-1}}$ | hard-coded (mo_constants) |
| Specific heat capacity of dry air | 1005.46 | $\mathrm{J\,kg^{-1}K^{-1}}$ | hard-coded (mo_constants) |
| Specific heat capacity of water vapour | 1869.46 | $\mathrm{J\,kg^{-1}K^{-1}}$ | hard-coded (mo_constants) |
| Specific heat capacity of sea ice | 2090 | $\mathrm{J\,kg^{-1}K^{-1}}$ | hard-coded (sicetemp) |
| Specific heat capacity of sea water | 4098 | $\mathrm{J\,kg^{-1}K^{-1}}$ | default (initcon/*simcons*) |
| Specific heat capacity of freshwater | 4186.84 | $\mathrm{J\,kg^{-1}K^{-1}}$ | hard-coded (mo_constants) |
| Reference density of sea ice | 910 | $\mathrm{kg\,m^{-3}}$ | default (sicetemp+initcon/*simcons*) |
| Reference density of snow | 300 | $\mathrm{kg\,m^{-3}}$ | mixed[b] (sicetemp/*simcons*) |
| Reference density of sea water | 1025 | $\mathrm{kg\,m^{-3}}$ | mixed[b] (sicetemp/*simcons*) |
| Reference density of freshwater | 1000 | $\mathrm{kg\,m^{-3}}$ | hard-coded (mo_constants+growth) |
| Specific latent heat of fusion | $3.32{\cdot}10^5$ | $\mathrm{J\,kg^{-1}}$ | default (initcon/*simcons*) |
| Specific latent heat of vaporization | $2.5008{\cdot}10^6$ | $\mathrm{J\,kg^{-1}}$ | hard-coded (mo_constants) |
| Specific latent heat of sublimation | $2.8345{\cdot}10^6$ | $\mathrm{J\,kg^{-1}}$ | hard-coded (mo_constants) |
| Thermal conductivity of sea ice | 2.1656 | $\mathrm{W\,m^{-1}K^{-1}}$ | default (sicetemp+initcon/*simcons*) |
| Thermal conductivity of snow | 0.31 | $\mathrm{W\,m^{-1}K^{-1}}$ | default (sicetemp+initcon/*simcons*) |
| Adaptation time of mixed layer | 86400 | s | namelist (*awisim*) |
| Minimum thickness for lateral freezing | 0.3 | m | hard-coded (growth) |
| Maximum thickness for lateral freezing | 0.8 | m | hard-coded (growth) |
| Ice area melting constant | 0.5 | | hard-coded (growth) |
| Freezing temperature of freshwater | 273.15 | K | hard-coded (mo_constants+growth) |
| Ice strength parameter | 30000 | $\mathrm{N\,m^{-2}}$ | namelist (*awisim*) |
| Ice strength decay constant | 20 | | namelist (*simcons*) |
| Eccentricity w.r.t. the ice rheology | 2 | | default (setice/*awisim*) |
| Ice-ocean drag coefficient | 0.0055 | | namelist (*simcons*) |
| Oceanic vertical diffusion coefficient | 0 | $\mathrm{m^2 s^{-1}}$ | namelist (*mixing*) |
| Oceanic vertical viscosity coefficient | 0.001 | $\mathrm{m^2 s^{-1}}$ | namelist (*mixing*) |
| Albedo of frozen snow | 0.84 | | hard-coded (su_albedo) |
| Albedo of melting snow | 0.77 | | hard-coded (su_albedo) |
| Albedo of frozen ice | 0.57 | | hard-coded (mo_albedo) |
| Albedo of melting ice | 0.51 | | hard-coded (su_albedo) |
| Albedo of melt ponds (maximum) | 0.36 | | hard-coded (mo_albedo) |
| Albedo of melt ponds (minimum) | 0.16 | | hard-coded (mo_albedo) |
| Albedo of sea water | 0.10 | | hard-coded (mo_albedo+initcon) |
| Snow thickness for 75% snow cover | 0.03 | m | hard-coded (albedos) |
| Maximum snow cover fraction | 0.99 | | hard-coded (albedos) |
| Maximum melt pond fraction | 0.22 | | hard-coded (mo_albedo) |

[a]If two subroutines are given, they refer to the respective subroutines in HIRHAM5 (first subroutine) and NAOSIM (second subroutine). If one or two subroutine(s) and one namelist are given, the constant is preset in the subroutine(s), but can be changed by a corresponding namelist parameter.

[b]The term "mixed" indicates that the constant is hard-coded in HIRHAM5 and set to the same value via namelist in NAOSIM.

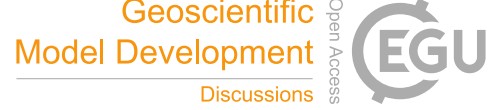



**Table B2.** Namelist parameters of the ECHAM5 physics as set for the base configuration of HN2.0.

| Namelist | Parameter | Purpose |
|---|---|---|
| physctl | lphys = true | Parameterize diabatic processes |
| | lrad = true | Use radiation scheme |
| | lvdiff = true | Allow vertical diffusion |
| | lcond = true | Use large-scale condensation scheme |
| | lcover = true | Use prognostic cloud cover scheme |
| | lconv = true | Allow convection |
| | lmfpen = true | Allow penetrative convection |
| | lgwdrag = false | Do not use gravity wave drag scheme |
| | lsurf = true | Allow surface exchanges |
| | lice = true | Calculate sea-ice temperature |
| | iconv = 1 | Use convection scheme by Nordeng (1994) |
| radctl | io3 = 2 | Use spectral ozone climatology |

**Table B3.** Preprocessor definitions used to build the two components of HN2.0.

| Symbol name | Component | Purpose |
|---|---|---|
| CPL_NAOSIM | HIRHAM5 | Run HIRHAM5 in coupled mode with NAOSIM |
| FLP_64B | HIRHAM5 | Use 64 bit floating point precision |
| BLAS | HIRHAM5 | Use routines from the BLAS library |
| SCOPY=dcopy | HIRHAM5 | Use DCOPY for vector copy operations |
| SGEMM=dgemm | HIRHAM5 | Use DGEMM for matrix-matrix operations |
| MPI_SRC | HIRHAM5 | Use MPI for the inter-process communication |
| GC | HIRHAM5 | Use the General Communication primitives package |
| MPI_REAL=MPI_DOUBLE_PRECISION | HIRHAM5 | Ensure 64 bit floating point communication |
| ECHAM5_PHYS | HIRHAM5 | Use ECHAM5 physics instead of HIRLAM physics |
| TOMPKINS_CLOUD_OPTION | HIRHAM5 | Use modified cloud parameterization by Klaus et al. (2016) |
| NEW_ALBEDO_FRACTIONS | HIRHAM5 | Use revised surface fractions in the albedo scheme |
| NOMPI | HIRHAM5 | Decomposition of ECHAM5 is controlled by HIRLAM |
| CPL_HIRHAM | NAOSIM | Run NAOSIM in coupled mode with HIRHAM5 |
| NEW_ICE_THERMO | NAOSIM | Use the flux-separating thermodynamic ice growth scheme |
| REDUCED_OUTPUT | NAOSIM | Only a reduced number of fields are stored in output files |





*Author contributions.* AR and KD led the development of HIRHAM5; and CK and RG led the development of NAOSIM. WD performed the coupling of HIRHAM5 and NAOSIM and carried out the coupled model experiments. WD prepared the manuscript with contributions from all co-authors.

*Competing interests.* The authors declare that no competing interests are present.

5  *Acknowledgements.* This work was supported by the SFB/TR 172 "ArctiC Amplification: Climate Relevant Atmospheric and SurfaCe Processes, and Feedback Mechanisms (AC)³" in project D03 funded by the German Research Foundation (DFG). We thank the European Centre for Medium-Range Weather Forecasts for providing the ERA-Interim reanalysis data. Special thanks goes to Moritz Hanke from the German Climate Computing Centre (DKRZ) for his great assistance in implementing the coupling software. Finally, we thank Gunnar Spreen, Ines Hebestadt, and Benjamin Segger for support in data preparation and netCDF output according to CF (Climate and Forecast) conventions.

10 The HIRLAM System was developed by the HIRLAM Programme group, a co-operative programme of the national weather services in Denmark, Estonia, Finland, Iceland, Ireland, Lithuania, the Netherlands, Norway, Spain, and Sweden.





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
