# Peer review of "HIRHAM–NAOSIM 2.0: The upgraded version of the coupled regional atmosphere-ocean-sea ice model for Arctic climate studies"

_Geoscientific Model Development, 2018_

## Referee Comment (RC1) · Anonymous Referee #1 · 21 Dec 2018

General comments:

This paper presents an upgraded atmosphere-ocean-sea-ice coupled model version for Arctic climate studies. This coupled configuration combines some new atmospheric parameterisations, an increased horizontal and vertical spatial resolution for all components and the use of a new coupler software. Then it assesses the atmospheric and the sea-ice components using an ensemble of 10 hindcast simulations over 1979-2016. This paper is well structured allowing an easy read; it also shows both improvements and (robust) weaknesses and proposes some clues to explore for solving them. So I think it suits the aims of GMD publication. This new coupled configuration has required,

without doubt, an important work that would deserve to be highlighted in adding an analysis of the oceanic component. I have a few comments and questions below that, hopefully, will help to improve the paper.

Major comment:

My only major comment concerns the oceanic component. Although, in the paper, it is mentioned that (pg 12, lines 1- 2) 'Sea ice is an important indicator for the overall performance of a coupled Arctic climate model, because it depends on atmospheric and oceanic processes to a similarly large extent.', the oceanic component is not assessed at all and I think this aspect is missing.

Therefore, I will suggest the authors to assess the oceanic component in both HN1.2 and HN2.0. They could look at the paper of Ilicak et al. (2016) in which the Arctic ocean has been assessed in a set of ocean-ice simulations performed with the same inter annual CORE surface forcing for a models inter-comparison exercise. Many diagnostics are proposed and could be reproduced here but I think their Figure 7 could be of interest. Indeed it represents the mean vertical temperature and salinity profiles in the Nansen and the Canadian basins and compare them to the PHC3.0 climatology. This will give an interesting Arctic ocean vertical stratification state in the two coupled version models as the potential discrepancies compare to the climatology.

The Figure 5 of the present paper shows that the initial ensemble mean sea-ice state in 1979 is different between HN1.2 and HN2; it might be worth to show how is the ocean ensemble mean vertical stratification in 1979 as well. As mentioned by the authors, the spin-up protocol is not the same between HN1.2 and HN2.0, meaning that ocean/sea-ice spin-up time length has been built differently. This can lead to a more or less marked ocean drift in the two ensemble HN1.2 and HN2.0 which might be useful to identify. I think this will add value to the paper in giving a complete view of the three physical components especially in a climate study framework.

The reference previously mentioned: Mehmet Ilicak, Helge Drange, Qiang Wang, Rüdiger Gerdes, Yevgeny Aksenov, David Bailey, Mats Bentsen, Arne Biastoch, Alexandra Bozec, Claus Böning, Christophe Cassou, Eric Chassignet, Andrew C. Coward, Beth Curry, Gokhan Danabasoglu, Sergey Danilov, Elodie Fernandez, Pier Giuseppe Fogli, Yosuke Fujii, Stephen M. Griffies, Doroteaciro Iovino, Alexandra Jahn, Thomas Jung, William G. Large, Craig Lee, Camille Lique, Jianhua Lu, Simona Masina, A.J. George Nurser, Christina Roth, David Salas y Mélia, Bonita L. Samuels, Paul Spence, Hiroyuki Tsujino, Sophie Valcke, Aurore Voldoire, Xuezhu Wang, Steve G. Yeager. An assessment of the Arctic Ocean in a suite of interannual CORE-II simulations. Part III: Hydrography and fluxes, Ocean Modeling, Volume 100, 2016, Pages 141-161, ISSN 1463-5003, https://doi.org/10.1016/j.ocemod.2016.02.004.

Minor comments:

1- Introduction:

pg 1, from line 10 to 15: the term feedbacks is never detailed, give few examples on major well known feedbacks even not fully understood ?

pg 1, from line 14 to 16 : 'Since these feedbacks …...' indicating that model upgrades are still needed. This sentence is not clear to me and might be reformulate

2- Description of HN2.0:

2-1-3 Boundary forcing:

pg 5, lines 10-11, to make the text easier to read, maybe just give only the lower boundary conditions of HIRHAM5.0 for the coupled configuration and not the stand alone case

2-2-4 Domain and discretization:

pg 7, line 15: '...but the model tends to crash every now and then...' I don't understand 'every now' here, need to be changed

pg 7, line 17 reformulate the sentence: ' ...the here presented simulations were running

" . . . by something like " . . .our simulations were running .. '

2-2-5 Boundary forcing:

pg 7, line 22: it might be worth to mention what is the lateral dynamical forcing in complement to the Levitus climatology T/S restoring

pg 7, line 27-29: as for the atmospheric component and to simplify, it might be helpful to focus only on the coupled and non coupled one boundary forcing. The stand alone mode doesn't help the reader

pg 8, line 3: as previously mentioned, talking about the stand alone mode is useless, focus only on the coupled and non coupled domain is enough. the key message is: in non coupled area, the atmospheric forcing are computing using NAOSIM bulk formula based on ERAI state variables such as wind components, clod, precipitation . . ..etc while over the coupled area the atmospheric fluxes are calculated by HIRHAM5 and transmitted to NAOSIM It might be interesting to mention the bulk formula type used in NAOSIM.

2-2-5 Technical modifications:

pg 8, line 15: the elapse time required to simulate 1 calendar year as the number of processors dedicated to both HIRHAM5 and NAOSIM is an interesting information that would be nice to mention here. The parallelization relies on MPI, OpenMP, hybrid MPI-OpenMP ?

2-3 Coupling procedure:

p8: it might be worth to precise the authors motivations for using YAC in NH2.0. What was the coupler used for HN1.2 ?

3 - Evaluation of the base configuration:

pg 10 line 13 & pg 11 lines 1-3: '. . .the local greenhouse effect is underestimated in the curent HN2.0 simulations...': does it means that air temperatures are also underestimated within the domain ? Details from Graham et al. (2017) (their Fig. 4.e) will be interesting to mention there. Furthermore, is this specificity (underestimation of local greenhouse effect) is also present in the HN1.2 version used here for the comparison ? This could be valuable to mention it.

3.1 Ensemble simulation setup:

pg 11, line 13: it would be helpful for the reader to get a clear information on the spin-up time length for both ocean-sea-ice components used as initial state for each 10 hindcast, i.e. spin-up range [33 - 42] years for HN2.0 if I am right and for HN1.2?

3.2 Data for model evaluation:

pg 11, line 26: the HN1.2 ensemble covers the period from 1979 to 2014, this period should be the one retained for the comparison with observations and with HN2.0 instead having 3 different periods, i.e. 1979-2016 for HN2.0 and 1979-2015 for satellite data. It will allow simplification.

pg 11, line 27-28: Differences in the simulation results between the ensembles of HN1.2 and HN2.0 thus indicate changes in the model performance. This sentence is a Âń shortcut Âż and is not convincing to me. I will agree if the ocean/ice ensemble setup for the spin-up was the same between the two ensemble which is not the case here. So the differences may not be only due to the changes in the model performance or I do not understand what the authors want to express.

pg 12: Figure 3: the mean seasonal cycle of sea-ice variables should be computed over the same common period between satellite data and the 2 ensemble hindcats as previously mentioned. Furthermore, the shaded colors are not visible enough, make them darker. For data voids around the North Pole, why not removing data from models instead filled gap with distance-weighted averages, it would make sense to do it rather than extrapolating for the comparison ?

3.3 - Sea-ice climatology:

pg 13, line 9: '. . . from january to august . . ..', word correction

pg 14, lines 5-6: the authors suggest the potential effect of lateral freezing reference value to explain the overestimation of both the sea-ice extent and area in HN2.0. How behave the seasonal cycle of air temperatures over the Arctic in HN2.0 compare to ERAI and HN1.2 ? I guess they could be lower leading to sea-ice formation combined to low value for the lateral freezing ?

3.4 - Sea-ice trends and variability:

pg 14, line 14-15: about the underestimated downward trend in sea-ice, it might be worth to detail here the implications of : * using a climatological boundary forcing for the oceanic component: do the authors think about ocean drift effect ? * constant greenhouse gases in HN2.0 and HN1.2: is it related to the air temperatures trend ? Add a comment (Figure 5) on the fact that the relatively weak ensemble mean sea-ice volume for both HN1.2 and HN2.0 in 1979 against PIOMAS (keeping in mind that PIOMAS is not an observed data set) might also explain this sea-ice underestimated trend if true ? And that this trend might be also lowered by weaker air temperatures in HN1.2 and HN2.0 (compared to ERAI) so limiting sea-ice melt process ?

3.5 - Near-surface air temperatures:

3.5.1 - Winter:

pg 17, line 8: '..be associated with an approximately by 10 % underestimated sea-ice concentration..'. Sentence to reformulate.

pg 18, Figure 7: I think that plotting 2-m air temperature differences against the ERA-interim data will clearly help the reader to catch spatial structures and also locations described in the text. Leave the ERA-interim air temperature field as it is and show differences for both HN1.2 and HN2.0

pg 20, line 12: mention the web site for the coupler YAC as it is done for the model source code ?

---

## Referee Comment (RC2) · Anonymous Referee #2 · 28 Dec 2018

The paper presents a new version of the regional coupled model HIRHAM-NAOSIM. The predominate changes are physical improvements in the atmospheric component, increased resolutions, new coupling system and computational efficiency. Results from a set of experiments are outlined for the atmospheric and sea ice components. It is important to document these models as they develop, and thus there are good reasons for GMD to want to publish this paper. However, there are a number of areas in which the manuscript needs to be improved. The paper is well written but could benefit from some clarifications. I recommend the paper is published subject to major revisions.

Major comments: 1. The authors highlight the importance of properly representing

feedbacks between atmosphere, sea ice and ocean in studying climate. They mention only the Arctic amplification as example, that is largely influenced by complex teleconnections with lower latitudes. I would suggest to include a description of limitations of regional modelling in climate studies, together with the advantages (lines 17-23 pg 1). A regional model covering the Arctic region does not have to adequately reproduce the overturning circulation that largely impact the Arctic properties, but the choice of the prescribed boundary conditions is crucial. Here an old and low-resolution climatological data set is used. Add a comment on that, too.

2. The ocean component is a key part of the climate system and largely affects the sea ice model performances. First, I am surprised that a very old version of MOM is used. I suggest to justify this choice and include a more complete description of the ocean component and its setup in the paragraph 2.2. Then, the ocean results are totally missing in the paper - a proper analysis and validation has to be included. Diagnostics from Ilicak et al 2016 (https://doi.org/10.1016/j.ocemod.2016.02.004) and Uotila et al 2018 (https://doi.org/10.1007/s00382-018-4242-z) can be followed.

3. More details on the HN1.2 runs are needed for a clearer understanding of the intercomparison results

4. The manuscript would largely benefit from a detailed analysis of the possible changes in the air-ocean-sea ice interactions

Minor comments:

Section 2.1.2 Pg 4, line 4: 0.25degree corresponds to approximately 27km at the equator. Is this the nominal resolution of the atmospheric grid?

Section 2.1.3 Pg 5, line 10: provide a clearer distinction between the setting of standalone mode and the coupled mode Pg 5, lines 11-12: what are the implications of different conditions applied to uncovered sea points?

Section 2.2 Pg 5: I suggest to add details on relevant differences between HRM and

FRM

Section 2.2.1 Pg 5: more details on the ocean components are needed, parameterizations, mesh, bathymetry, etc.

Section 2.2.4 Pg 7, lines 1-2: the description of Arakawa B grid corresponds to the description of C grid in 2.1.2 Pg 7, lines 15-19: a more accurate description of numerical instability is needed. How are the authors sure that the model crashes are all ascribed to the choice of the mixing time step? Pg 7, lines 27-29: as for the atmospheric component, provide a clear distinction between stand-alone and coupled setting, and comment the possible implication of different forcing in coupled and uncoupled sectors of the domain

Section 2.2.6 Pg 8, lines 11-15: add a description on the new parallelization since this is one of the major improvements to the system. Which method has been used? Which component of the system is affected? Only NAOSIM is mentioned. How the impact on stand-alone simulations compares with coupled simulations?

Section 2.3 Pg 8, line 16: motivate the choice of YAC version 1.2, also compared to the previous coupler.

Section 3 Pg 10, line 10: the namelist of ocean and sea ice parameters is missing in the manuscript

Section 3.1 Pg 11, lines 6-14: this is unclear, I suggest to rewrite the entire paragraph adding precise information on the spin-up time for NAOSIM and HIRHAM, and for the coupled HN2.0 runs. The HN1.2 runs follow the same spin-up strategy? "The simulations were driven by ERAI data" refers to the coupled runs? If so, I suggest to reword, "driven" generally is for an ocean-sea ice simulation forced by atmospheric reanalysis.

Section 3.2 Pg 11, line 20: how robust is the validation against ERA Interim since it has been used for the initialization? Why not to use independent data? Pg 11, line 22: reword "quasi realistic" Pg 11, lines 27-28: clear statements on the differences and similarities between HN2.0 and HN1.2 would help, in addition to Graham et al 2017. I do not understand the sentence "Differences in the simulation results . . . indicate changes in the model performance". There are many differences between the 2 versions and probably between model set-up and spin-up. Please clarify.

Section 3.3 Pg 12, lines 1-3: a good representation of sea ice properties does not guarantee a good representation of ocean and atmospheric fields. I think that assessing the quality of ocean/ atmosphere components would largely improve the manuscript. For example, how does the increased ocean resolution impact the ocean circulation and water properties in the Arctic and consequently the sea ice? Pg 12, Figure 3. I suggest to compute the seasonal cycle over the same period for the three products. Pg 13, lines 4-5: where the thicker sea ice in HN2.0 comes from? The amplitude of the melting season is similar from the area/extent seasonal cycle. How different are the sea ice properties (concentration, thickness, temperature, etc.) in the initialization fields? Explain the different amplitude of the volume seasonal cycle between the 2 model versions. Pg 13, line 7: please define "relatively thick and thin ice". Maybe a distinction between pack ice and marginal zone ice may help. Which mechanisms (dynamics, thermodynamics, both) improve the thickness representation in HN2.0? Is the ice drift similar in the two models? Pg 13, line 9: change Januar to January Pg 14, lines 5-6: could this differences in the growing season also be related to differences in the ocean and atmosphere between the two models? Are similar are, for example, air temperature and sea surface temperature in HN1.2 and HN2.0? Which one is the main driver of ice growth in the model?

Section 3.4 Pg 14, line 12: the agreement between PIOMAS and HN2.0 is "reasonable" only in March, the variability in September is not captured in the most recent years. For instance, the 2007 and 2012 minima are not reproduced. Given the modelled trend and the variability, I would not call "agreement" the overlap between curves. Why does sea ice extent in NH2.0 (mainly in March) present weaker inter-annual variability? Pg 14, line 16: ". . .trend in sea-ice volume . . . can thus only arise from large-scale atmospheric

changes". I do not believe that is true. What are the differences in the two oceans? Is the variability of air temperature the same in the models? How are the feedbacks between the two components affected by the new coupler? Pg 14, lines 29-34: I did not understand the message within those lines. Please, rephrase.

Section 3.5.1 Pg 16, line 5: Kelvin in the text and Celsius in Fig.7. Use the same.

Section 3.5.2 Pg 17, line 5: 0 degree is the freezing temperature of freshwater (no salt in it). This is not the case for the Arctic ocean upper layer. Rephrase. Pg 17, line 7-8: about the reason of different summer temperature, how different are the heat fluxes between ocean and atmosphere in the two models? Is the air/ocean poleward heat transport the same? Then, rephrase "with an approximately by 10% underestimated sea ice concentration..." Pg 18, Figure 7: it might be more useful for the reader to have directly the plots of the differences HN2.0 – HN1.2 and HN2.0 – ERA Interim. It would help to add a contour indicating to the Arctic water freezing point. Pg 18, line 2: does the sea ice model include a melt-pond scheme? If so, which one? Was the same in HN1.2? Pg 18: maybe a comment on differences in solid and liquid precipitation between the 2 models and the comparison with ERA Interim might be helpful

Section 4 Title of section 4 "Conclusions: I do not detect so many conclusions or discussion on the model performances, more future work. It would be nice to add some conclusions drawn from the model results; alternatively rename Section 4 to "Conclusions and Future work". Pg 19, line 11-17: this study might also suggest that the physical core of the regional model components needs larger improvements. From those lines, a question arises whether the manuscript should include a better tuning and so better results. I would suggest to reformulate. Pg 20, line 3: how are the snow and ice albedo defined in HN1.2? Pg 20, Code availability: add the link to Max Planck InstituteÂăwebpage on YAC

I do not think that Table A1 and table A2 are necessary. For example, for the ocean depth, it might be enough adding something like: the layer thickness is 10m from the

surface to 215m and then increases up to about 350m at the bottom.

---

## Referee Comment (RC3) · Anonymous Referee #3 · 22 Jan 2019

This paper presents an upgraded version of a regional climate model covering a pan-Arctic region. The main difference is the use of a new atmospheric modelling component. The modifications to the model are described and a basic evaluation of sea ice and surface air temperature are presented. The main aim is to provide a reference for this new configuration to be used in further regional climate applications. As such, publication in Geoscientific Model Development seems appropriate. However, I find that the model description could be more clear and that the evaluation, focusing only on basic properties of the sea ice and near-surface temperature, is too limited. I recommend this paper for publication with major revisions. General and specific comments are provided below.

General Comments:

1. Since the main purpose of this paper is to document changes to the coupled model, it is important that this description be clear and thorough. However, the authors fail to clearly present the differences between the new version HN2.0 and the previous version (HN1.x). The main change is the use of a new atmospheric model, which itself is built on two previsouly described components (HIRLAM7 and ECHAM5.4). The model description is often quite difficult to follow as the authors intermingle modifications with respect to HN1.x with modifications to HIRLAM7 and ECHAM5.4. I recommend this section be rewritten to make these differences clear. In particular, if the aim here is to document the differences between HN2.0 and HN1.2 than these should be outlined in detail, and not rely on previous publications of HIRLAM7 and ECHAM5.4. Without a clear description of direct differences between HN1.2 and HN2.0 it is difficult to interpret the results of the model evaluation presented in Section 3.

2. While I agree that sea ice is an important indicator of overall model performance, a reference paper such as this is more useful when a broader presentation of model performance is outlined. Given the large changes in the atmospheric component I would have expected to see a more detailed description of characteristics of the modelled atmosphere.

Specific comments:

1. Pg1, line 10: "allow to simulate". Please rephrase, perhaps "allow one to simulate" or similar. 2. Section 1, para 3: It would be helpful to explain the motivations for upgrading the atmospheric component and any particular deficiencies that it is aiming to overcome. Also, the choice for the particular components chose for HN2.0 could be justified (ie HIRLAM7 and ECHAM5). 3. Pg2, line 18: Regardless if they have been described in reference manuals, if the aim of this paper is to document the new model version than a description of model components should be provided here. 4. Pg. 3, line 6: spelling error, should be "aerosol" 5. Pg. 3, line 8-9: "The most important

modification" from what? From HN1.2 or ECHAM5? Mixing these up makes the text difficult to follow. Also statements like "for the most part" are vague and should be avoided. Rather, explain what has been changed and what hasn't. 6. Pg. 3, line 17-18: "...attenuated such that at least 25%...". This sentence is quite difficult to follow. If this is the most important modification then it would be worth including the equation and describing this properly. Also it seems it may be relevant for the sea ice results presented in Section 3 (?). 7. Pg. 3, Line 22: "The second modification...". From what? ECHAM or HN1.2? 8. Pg. 4, line 13: semi-Lagrangian advection schemes are known to have conservation issues when used at high CFL number. For a weather model this is usually not a problem, but for a regional climate model this could affect the results. A discussion of this issue and the extent to which HN2.0 is conservative should be included, perhaps with some demonstration of applicable CFL numbers. 9. Pg. 5, line 17: "fine resolution" and "high-resolution" are not very useful. Please include a more precise indication of model resolution. Also, on pg2, line 16 it is noted that the ocean component is "largely the same". If the model configuration has completely changed this statement is not accurate. Moreover, simply stating that the difference in model configuration is described in Fieg et al (2010) is not sufficient. At least a brief description should be provided here as well. 10. Section 2.2.3: How is this different from HN1.x? 11. Section 2.2.4: How is this different from HN1.x? 12. Pg. 7, line 29: Is there any blending used when going from HIRHAM5 forcing to ERAI? 13. Pg. 8, line 1: "standard bulk formulas". Please describe. Are these the same bulk formulas used by HIRLAM when coupled? 14. Pg. 8, line 8: Is this the only difference in how fluxes are calculated? For example, are surface roughnesses and boundary layer stability all treated the same? 15. Pg. 11, line 10: The use of ice-ocean fields from Januaries 1991 to 2000 seems a rather odd choice. Some explanation should be provided. Also, since thickness over this period were thinner than for the earlier period, please describe any impact on mean sea ice results (i.e. due you see any spin up effects? Is there any change is ensemble spread from year1 to year 20+? 16. Pg. 11, line 27: If the main comparison presented in this paper is against this HN1.2 ensemble, than an

explanation of how the setup differs should be given. 17. Fig. 3: (middle). It would be helpful to include PIOMAS here as well to be able to differentiate spatial differences (area) from thickness contributions to total volume. 18. Pg. 14, line 7: "had been resulted" change to "...resulted..." or similar. 19. Pg. 14, line 19: "observation-like" is a bit of an unusual term. Perhaps change this to "reference" 20. Pg. 14, line 32-33: The inability to simulate extrema is not necessary just a matter of model internal variability though as many key processes are missing (e.g. wave-ice interactions which played an important role in the 2012 minimum that is used as an example). It would be good to note this limitation in simulating extremes and comment on the degree to which this may be important for simulations with this regional climate model. Since HN2.0 has a higher resolution ocean-ice model, does this affect extremes? 21. Pg. 17, line 5-10. It would be helpful to show some additional diagnostics here associated with albedo and surface heat fluxes to understand better the source of these differences. 22. Pg. 18, line 3: If there is increased melting from the ocean, may this also be related to changes in ocean transports. A higher resolution ocean configuration may allow more Atlantic water to enter the Arctic via Fram strait and the Barents Sea. Some comment/validation of this would be helpful to understand how the behavior of HN2.0 differs from HN1.2. 23. Pg. 19, line2: "...be solved until now" change to "...as of now".

---

## Author Comment (AC1) · 18 Mar 2019

**Author Comments to the Comments of Referee #1**

Response to the major comment

The referee's only major comment concerns the assessment of the ocean component. We absolutely agree that a detailed evaluation of the ocean component would be a valuable addition to the presented more technical description of the model development. However, there are a couple of reasons why we decided to focus in this paper

only on a base evaluation with respect to sea ice as the communicator between atmosphere and ocean:

- 1. The two component models HIRHAM5 and NAOSIM have already been used in previous studies that also include their evaluation. Since the setups of the component models have been left unchanged for the present coupled model version as far as possible, the primary task is to demonstrate that the new coupling procedure with the aid of YAC is technically working properly and that interactions between the component models are actually represented in an acceptable way.
- 2. The decision trying to publish the upgraded version of the coupled model in GMD as "Development and technical paper" has been made in order to allow a more detailed description of technical details from which in particular model users may benefit, but also other model developers who want to couple different model components. For this very reason, a more detailed model evaluation with a geoscientific focus was not intended and is beyond the scope of a development and technical paper in GMD.
- 3. A full evaluation of the entire model, not only with respect to the ocean component, but also with respect to the not less complex atmosphere component, would be so substantial that one or more stand-alone papers are required or at least highly recommended for a detailed evaluation of different aspects of the Arctic climate system. Some of these aspects, for instance sea-ice drift, Atlantic water inflow, and atmospheric cyclones, are already subject of our current research and will likely result in follow-up papers in pure scientific journals.
- 4. The current base configuration of the coupled model is certainly not the final model configuration, as already noted in the manuscript. A comparison with other fine-tuned coupled models, or even with pure ocean-sea ice models which use CORE-II interannual forcing and do not comprise feedback processes between
atmosphere and ocean-sea ice components, makes little sense at the present stage and should better be postponed until the development process towards an improved configuration will have been completed. However, the current base configuration of the upgraded coupled model already performs better than the old model version, indicating that interactions between the model components have clearly improved in spite of imperfect model configuration.

5. A further and more technical reason why we did not take into consideration to evaluate the deep ocean is the simple fact that such data are not available from the old model version (HN1.2). A comparison of vertical temperature and salinity profiles between HN2.0 and HN1.2 is therefore impossible.

As a reasonable compromise, because two of the three referees requested to include evaluation of the ocean component, we have incorporated a new subsection (section 3.5) in which the upper ocean temperature of HN1.2 and HN2.0 is compared with the PHC3.0 climatology. The upper ocean temperature was selected due to its direct influence on the sea-ice conditions. It also provides additional insight into the different bias structure of HN1.2 and HN2.0 and might be a valuable supplement to the discussion of model improvements that are directly related to the coupling.

Because we have kept the focus mainly on a base evaluation with respect to sea ice as the communicator between atmosphere and ocean, and in order to clarify what the reader may (and may not) expect from the paper, we have added the following sentence to the abstract: "The evaluation focuses mainly on sea ice as the communicator between atmosphere and ocean." We have also added a brief outlook in the Conclusions section that more detailed evaluation of the model is postponed until the development process towards an improved configuration will have been completed and will be subject of follow-up studies.

Another comment made by the referee with respect to the ocean component was the initial ensemble mean sea-ice state in 1979. The referee notes that this state is dif-

GMDD
ferent between HN1.2 and HN2.0 and argues that this difference might be related to a more or less marked ocean drift in the two ensembles HN1.2 and HN2.0. Actually, the difference in the ensembles' mean sea-ice state in 1979 is an expression of different steady states due to differences in the model physics. Even though the initial states of the two ensembles have been selected on a slightly different basis, this basis was in either case an already steady-state simulation of the respective coupled model version with its specific model physics. This setup for building an ensemble of coupled model simulations has already been used in previous studies (e.g., Dorn et al., 2012) and ensures an minimization of internal model drift due to the use of initial conditions from the steady state of a coupled spin-up run with the identical model version as used for the ensemble simulations themselves. Therefore, ocean drift is not a major issue in this case.

Clear information that all HN2.0 ensemble members were initialized with ocean-ice conditions from the steady state of the specific model configuration has now been added to the description of the ensemble simulation setup. Later, when we mention the HN1.2 ensemble for the first time, we have now pointed explicitly to the comparability of the two ensemble setups from a scientific point of view, even if they differ technically.

Response to the minor comments

In the following, a point-by-point response to the referee's minor comments is given in the sequence of comment (C) and answer (A).

C: pg 1, from line 10 to 15: the term feedbacks is never detailed, give few examples on major well known feedbacks even not fully understood ?

A: We have added two examples of feedbacks, the well-known ice-albedo feedback and the not fully understood feedback between sea ice and cyclones.

C: pg 1, from line 14 to 16 : 'Since these feedbacks . . ...' indicating that model
upgrades are still needed. This sentence is not clear to me and might be reformulate

A: We have removed the whole sentence, because it is not absolutely necessary for the paper.

C: pg 5, lines 10-11, to make the text easier to read, maybe just give only the lower boundary conditions of HIRHAM5.0 for the coupled configuration and not the stand alone case

A: Information on HIRHAM5 stand-alone has been removed in this paragraph as suggested.

C: pg 7, line 15: '...but the model tends to crash every now and then. . .' I don't understand 'every now' here, need to be changed

A: The phrase "every now and then" was here used as a synonym for "from time to time" or "occasionally". We have replaced this phrase by the latter.

C: pg 7, line 17 reformulate the sentence: ' ...the here presented simulations were running "... by something like "...our simulations were running ...'

A: The term "the here presented simulations" has been replaced by "the HN2.0 simulations".

C: pg 7, line 22: it might be worth to mention what is the lateral dynamical forcing in complement to the Levitus climatology T/S restoring

A: The formulation of the open-boundary condition by Stevens (1991) used in NAOSIM does not require forcing data for solving the equations of motion due to the neglect of the nonlinear advection terms at the boundary and extrapolating from the boundary to outside the boundary for the horizontal diffusion terms. Thus, there is nothing to mention with respect to the lateral dynamical forcing.

C: pg 7, line 27-29: as for the atmospheric component and to simplify, it might be helpful to focus only on the coupled and non coupled one boundary forcing. The stand
alone mode doesn't help the reader

A: Information on NAOSIM stand-alone has been removed in this paragraph as suggested.

C: pg 8, line 3: as previously mentioned, talking about the stand alone mode is useless, focus only on the coupled and non coupled domain is enough. the key message is: in non coupled area, the atmospheric forcing are computing using NAOSIM bulk formula based on ERAI state variables such as wind components, clod, precipitation . . ..etc while over the coupled area the atmospheric fluxes are calculated by HIRHAM5 and transmitted to NAOSIM It might be interesting to mention the bulk formula type used in NAOSIM.

A: Information on the stand-alone mode has also been removed in this paragraph. The bulk formulas used in NAOSIM are based on the formulations for turbulent fluxes and shortwave radiation by Parkinson and Washington (1979) and for longwave radiation by Rosati and Miyakoda (1988). We have added this information to the paper.

C: pg 8, line 15: the elapse time required to simulate 1 calendar year as the number of processors dedicated to both HIRHAM5 and NAOSIM is an interesting information that would be nice to mention here. The parallelization relies on MPI, OpenMP, hybrid MPI-OpenMP ?

A: The paragraph has been extended by more specific information on the reparallelization. The number of processors allocated to HIRHAM5 and NAOSIM for coupled simulations and the corresponding elapsed time required to simulate one calendar year have been added.

C: *p8: it might be worth to precise the authors motivations for using YAC in NH2.0. What was the coupler used for HN1.2 ?*

A: A brief description of the coupling procedure in HN1.2 and earlier versions has been added to section 2.3, directly leading to the motivation for using YAC in HN2.0.
C: pg 10 line 13 & pg 11 lines 1-3: '. . .the local greenhouse effect is underestimated in the curent HN2.0 simulations...': does it means that air temperatures are also underestimated within the domain ? Details from Graham et al. (2017) (their Fig. 4.e) will be interesting to mention there. Furthermore, is this specificity (underestimation of local greenhouse effect) is also present in the HN1.2 version used here for the comparison ? This could be valuable to mention it.

A: Air temperatures are indeed a little too cold in HN2.0. This finding is already mentioned in section 3.5 when discussing the near-surface air temperatures. Figure 4e by Graham et al. (2017) show results from the old model version (HN1.2). A comparable analysis with HN2.0 has not yet been made and is definitely beyond the scope of the current paper. The greenhouse gases in HN1.2 were set to values representative for the year 1990 (the default values of the ECHAM4 parameterizations used in HN1.2). When explicitly pointing to an erroneous setting of greenhouse gases in the current HN2.0 simulations, it should be implicitly clear that this error does not appear in the HN1.2 simulations.

C: pg 11, line 13: it would be helpful for the reader to get a clear information on the spin-up time length for both ocean-sea-ice components used as initial state for each 10 hindcast, i.e. spin-up range [33 - 42] years for HN2.0 if I am right and for HN1.2?

A: From previous studies it is known that the coupled regional model needs a spin-up time of about 6–10 years to reach a quasi-stationary seasonal-cyclic state of equilibrium (Dorn et al., 2007). If the initial ice conditions are not far away from this state, the spin-up time will be even shorter. This result was found in simulations with HN1.1 and has been experimentally verified with HN2.0. Since the initial conditions of all ensemble simulations with both HN2.0 and HN1.2 were taken after more than 10 years of spin-up, they all represent the steady state of the respective model version. Consequently, the specific length of spin-up that exceeds the 10-year limit is completely irrelevant. Clear information that all ensemble members were initialized with ocean–ice conditions from the steady state of the specific model configuration has now been
added to the description of the ensemble simulation setup.

C: pg 11, line 26: the HN1.2 ensemble covers the period from 1979 to 2014, this period should be the one retained for the comparison with observations and with HN2.0 instead having 3 different periods, i.e. 1979-2016 for HN2.0 and 1979-2015 for satellite data. It will allow simplification.

A: We have recompiled Figure 3 and Figure 4 using data only for the period 1979–2014 in all cases. There are only minor, almost undetectable variances compared to the old figures so that adaptations of the text (beyond the figure captions) have not been needed.

C: pg 11, line 27-28: Differences in the simulation results between the ensembles of HN1.2 and HN2.0 thus indicate changes in the model performance. This sentence is a shortcut and is not convincing to me. I will agree if the ocean/ice ensemble setup for the spin-up was the same between the two ensemble which is not the case here. So the differences may not be only due to the changes in the model performance or I do not understand what the authors want to express.

A: As aforementioned, the ensemble setup of HN1.2 and HN2.0 is comparable from a scientific point of view, even if it differs technically. Therefore, the differences in the simulation results between the ensembles of HN1.2 and HN2.0 can surely be rated as indication of changes in the model performance. We have added the reference of the scientific comparability of the two ensembles before pointing to the changes in the model performance due to differences in the physical process descriptions" in order to express that we here do not refer to the technical or computational performance, which is clearly better in HN2.0, but is not expressed by the differences in the simulation results.

C: pg 12: Figure 3: the mean seasonal cycle of sea-ice variables should be computed over the same common period between satellite data and the 2 ensemble hindcats as previously mentioned. Furthermore, the shaded colors are not visible enough, make
them darker. For data voids around the North Pole, why not removing data from models instead filled gap with distance-weighted averages, it would make sense to do it rather than extrapolating for the comparison ?

A: As already noted, all data sets of the new Figure 3 refer now to the period 1979–2014. The shaded areas have been made darker (transparency changed from 50% to 80%). Data voids around the North Pole in the satellite data are still filled with distance-weighted averages. The area around the North Pole is the central region of the coupled model system and is year-round ice-covered and therefore important. The extrapolation of the satellite data using distance-weighted averages to fill data voids is an adequate method that leads to reasonable results and enables an unbiased comparison with the model results for the identical domain.

C: pg 13, line 9: '. . . from january to august . . ..', word correction

A: The word 'January' has been corrected.

C: pg 14, lines 5-6: the authors suggest the potential effect of lateral freezing reference value to explain the overestimation of both the sea-ice extent and area in HN2.0. How behave the seasonal cycle of air temperatures over the Arctic in HN2.0 compare to ERAI and HN1.2 ? I guess they could be lower leading to sea-ice formation combined to low value for the lateral freezing ?

A: The important role of the parameterization of lateral freezing in uncoupled and coupled models was already investigated in several previous studies (Fichefet and Morales Maqueda, 1997; Bitz and Lipscomb, 1999; Dorn et al., 2007; Wang et al., 2010; Mauritsen et al., 2012; Notz et al., 2013; Shi and Lohmann, 2017). The reference thickness for lateral freezing basically controls how fast open-water areas are freezing up when the surface energy balance is negative. A reduced value leads to faster freeze-up and reduces the heat transfer from the ocean to the atmosphere due to the insulating ice cover. One of the consequences are colder near-surface air temperatures, another consequence is decelerated ice thickness growth due to reduced heat loss to the atmoGMDD
sphere. Thinner ice, in turn, is associated with increased conductive heat flux through the ice which damps the near-surface cooling of the atmosphere, but cannot compensate for the differences in the ocean-atmosphere heat flux due to reduced open-water areas. Consequently, the colder near-surface air temperatures, which are bound to occur during the freezing season in the HN2.0 simulations, play only a minor part in the sea-ice formation process. This partial aspect of the complex interaction between ice growth and near-surface air temperatures represents a negative feedback mechanism. Concerning negative feedbacks, it is pretty difficult to adjust a model in a reasonable way. At least fine-tuning of the reference thickness for lateral freezing is one possibility to minimize biases in sea-ice thickness and concentration as well as in near-surface air temperatures for lateral freezing might be responsible for the sea-ice bias and represents one possibility to minimize the bias, we have added a brief discussion with reference to the study by Dorn et al. (2007) at the end of the paragraph in question.

C: pg 14, line 14-15: about the underestimated downward trend in sea-ice, it might be worth to detail here the implications of : \* using a climatological boundary forcing for the oceanic component: do the authors think about ocean drift effect ? \* constant greenhouse gases in HN2.0 and HN1.2: is it related to the air temperatures trend ? Add a comment (Figure 5) on the fact that the relatively weak ensemble mean seaice volume for both HN1.2 and HN2.0 in 1979 against PIOMAS (keeping in mind that PIOMAS is not an observed data set) might also explain this sea-ice underestimated trend if true ? And that this trend might be also lowered by weaker air temperatures in HN1.2 and HN2.0 (compared to ERAI) so limiting sea-ice melt process ?

A: The use of a climatological boundary forcing for the oceanic component is indeed one important obstacle for a realistic reproduction of the observed downward trend. Long-term changes in the oceanic heat transport, which might have contributed to the trend, cannot take effect in the current model configuration. An artificial ocean drift effect, which could counteract the downward trend, is not really detectable in the
model simulations. As already noted before, all model simulations started from steady states. Test simulations with more realistic greenhouse gases showed that the observed downward trend in sea ice and the related upward trend in air temperatures is still underestimated, even though a little less. Therefore, we suppose that the oceanic boundary forcing plays a similarly important role as the atmospheric boundary forcing, while the local greenhouse effect is a second-order effect. This supposition still needs to be proved by elaborated sensitivity experiments that are already scheduled. At least one question raised by the referee can definitely be answered: The relatively weak ensemble mean sea-ice volume for both HN1.2 and HN2.0 in 1979 against PIOMAS does not contribute in any way to the underestimated trend in sea ice. Simulations with higher initial ice volume show comparable underestimations. Also, the (local) air temperatures do not play a role for the underestimated sea-ice trend, because they do not represent forcing variables, but prognostic variables of the coupled model system. Local temperature changes and sea-ice changes are therefore closely linked to each other. The question should rather be how well the physical processes, which constitute this linkage, are simulated by the model? Concerning this matter, HN2.0 does a clearly better job than HN1.2 as can be seen from the lower biases in both sea ice and surface temperature.

C: pg 17, line 8: '..be associated with an approximately by 10 % underestimated seaice concentration.'. Sentence to reformulate.

A: The sentence has been reworded. The clause in question reads now "... be associated with the underestimation of sea-ice concentration by about 10%..."

C: pg 18, Figure 7: I think that plotting 2-m air temperature differences against the ERA-interim data will clearly help the reader to catch spatial structures and also locations described in the text. Leave the ERA-interim air temperature field as it is and show differences for both HN1.2 and HN2.0

A: We agree that it is easier to identify the differences between the simulations and

**GMDD**
ERA-Interim when showing difference plots. At the time when we thought about the selection of figures for the current paper, we already had difference plots in mind, but finally decided to include the plots of the temperature climatologies for reasons of consistency with Figure 4. This is of course not a compelling reason. Therefore, we followed the referee's suggestion and have replaced the subfigures of the HN1.2 and HN2.0 climatologies by their respective difference to ERA-Interim.

C: pg 20, line 12: mention the web site for the coupler YAC as it is done for the model source code ?

A: Information about access to the coupling software YAC has been added to the section 'Code availability'.

**References**

- Bitz, C. M. and Lipscomb, W. H.: An energy-conserving thermodynamic model of sea ice, J. Geophys. Res., 104, 15669–15677, 1999.
- Dorn, W., Dethloff, K., Rinke, A., Frickenhaus, S., Gerdes, R., Karcher, M., and Kauker, F.: Sensitivities and uncertainties in a coupled regional atmosphere–ocean–ice model with respect to the simulation of Arctic sea ice, J. Geophys. Res., 112, D10118, doi:10.1029/2006JD007814, 2007.
- Dorn, W., Dethloff, K., and Rinke, A.: Limitations of a coupled regional climate model in the reproduction of the observed Arctic sea-ice retreat, The Cryosphere, 6, 985–998, doi:10. 5194/tc-6-985-2012, 2012.
- Fichefet, T. and Morales Maqueda, M. A.: Sensitivity of a global sea ice model to the treatment of ice thermodynamics and dynamics, J. Geophys. Res., 102, 12609–12646, 1997.
- Graham, R. M., Rinke, A., Cohen, L., Hudson, S. R., Walden, V. P., Granskog, M. A., Dorn, W., Kayser, M., and Maturilli, M.: A comparison of the two Arctic atmospheric winter states observed during N-ICE2015 and SHEBA, J. Geophys. Res. Atmos., 122, 5716–5737, doi: 10.1002/2016JD025475, 2017.

Mauritsen, T., Stevens, B., Roeckner, E., Crueger, T., Esch, M., Giorgetta, M., Haak, H., Jung-
- claus, J., Klocke, D., Matei, D., Mikolajewicz, U., Notz, D., Pincus, R., Schmidt, H., and Tomassini, L.: Tuning the climate of a global model, J. Adv. Model. Earth Syst., 4, M00A01, doi:10.1029/2012MS000154, 2012.
- Notz, D., Haumann, F. A., Haak, H., Jungclaus, J. H., and Marotzke, J.: Arctic sea-ice evolution as modeled by Max Planck Institute for Meteorology's Earth system model, J. Adv. Model. Earth Syst., 5, 173–194, doi:10.1002/jame.20016, 2013.
- Parkinson, C. L. and Washington, W. M.: A large-scale numerical model of sea ice, J. Geophys. Res., 84, 311–337, 1979.
- Rosati, A. and Miyakoda, K.: A general circulation model for upper ocean simulation, J. Phys. Oceanogr., 18, 1601–1626, 1988.
- Shi, X. and Lohmann, G.: Sensitivity of open-water ice growth and ice concentration evolution in a coupled atmosphere-ocean-sea ice model, Dyn. Atmos. Ocean, 79, 10–30, doi:10.1016/j.dynatmoce.2017.05.003, 2017.
- Stevens, D. P.: The open boundary condition in the United Kingdom Fine-Resolution Antarctic Model, J. Phys. Oceanogr., 21, 1494–1499, 1991.
- Wang, Z., Lu, Y., Wright, D. G., and Dupont, F.: Sea ice sensitivity to the parameterisation of open water area, J. Oper. Oceanogr., 3, 3–9, doi:10.1080/1755876X.2010.11020113, 2010.

---

## Author Comment (AC2) · 18 Mar 2019

**Author Comments to the Comments of Referee #2**

Response to the major comments

In the following, a point-by-point response to the referee's major comments is given in the sequence of comment (C#) and answer (A#), where # refers to the numbering of the referee's major comments.

[Figure]

C1: *The authors highlight the importance of properly representing feedbacks between atmosphere, sea ice and ocean in studying climate. They mention only the Arctic amplification as example, that is largely influenced by complex teleconnections with lower latitudes. I would suggest to include a description of limitations of regional modelling in climate studies, together with the advantages (lines 17-23 pg 1). A regional model covering the Arctic region does not have to adequately reproduce the overturning circulation that largely impact the Arctic properties, but the choice of the prescribed boundary conditions is crucial. Here an old and low-resolution climatological data set is used. Add a comment on that, too.*

A1: Arctic amplification is mentioned as an example, because advanced knowledge about Arctic amplification is the overall objective of the project funding. The upgraded version of the coupled model system has been developed in order to provide contributions to this objective. Also, and this is the most important point, our proposed application of the regional model is not to simulate future climate changes as realistically as possible, but to simulate interaction processes between atmosphere, sea ice, and ocean in the Arctic, and to better understand feedback processes that contribute to Arctic amplification. Having this in mind, the constrained lateral boundary conditions are indeed a major advantage, since the model does not need to show reasonable performance in simulating global teleconnections. The overturning circulation is largely imposed by the ocean boundary conditions. Of course, variability in the overturning circulation is not captured using a climatological ocean boundary, but this limitation is irrelevant for the description of the coupled model system and its basic evaluation that clearly demonstrates that the new coupling procedure with the aid of YAC is technically working properly and that interactions between the component models are actually represented in an acceptable way. However, we agree that the choice of the lateral boundary forcing is an important issue. Accordingly, we have included statements about this limitation. In the revised version of the paper, we have mentioned that "the coupled regional climate model has to rely on the atmospheric and oceanic data used as lateral boundary forcing" and that "this is a particular issue for the lateral ocean boundary and
represents a limitation of coupled regional climate models in adequately reproducing observed climate changes." Later, when we mention the Levitus climatology for the first time, we have added a comment on the choice of this climatology: "The Levitus climatology was chosen to remain comparable to previous NAOSIM stand-alone and HN1.2 simulations." In the Conclusions section, we have now noted that "the model might also benefit from time-varying lateral ocean boundary forcing from ocean reanalyses which are envisaged to replace the Levitus climatology in the future."

C2: *The ocean component is a key part of the climate system and largely affects the sea ice model performances. First, I am surprised that a very old version of MOM is used. I suggest to justify this choice and include a more complete description of the ocean component and its setup in the paragraph 2.2. Then, the ocean results are totally missing in the paper - a proper analysis and validation has to be included. Diagnostics from Ilicak et al 2016 and Uotila et al 2018 can be followed.*

A2: We admit that the MOM-2-based model is an old model, but this version of MOM is an integral part of NAOSIM. Both component models, HIRHAM5 and NAOSIM, already existed in the present form before we started to coupled them. In the revised version of the paper, we have now included more details on the two component models that might be helpful to understanding the upgraded coupled model system without the requirement of reading secondary literature. This includes the improvement of section 2.2 on the NAOSIM description. NAOSIM has already been used in previous studies that also include their evaluation. These studies showed that the model is successful in simulating the ocean circulation in the Arctic (e.g., Drange et al., 2005; Karcher et al., 2012; Aksenov et al., 2016), Arctic Ocean freshwater content (Rabe et al., 2014), sea-ice concentration (e.g., Adams et al., 2011) and drift (e.g., Rozman et al., 2011), or the impact of cyclones on the sea ice in the Arctic Ocean (Kriegsmann and Brümmer, 2014). This justifies its usage. Furthermore, NAOSIM takes part in the international "Forum for Arctic Ocean Modeling and Observational Synthesis (FAMOS)" project, where state-of-the-art ice–ocean models are intercompared and evaluated. There, NAOSIM does not

stand out as an outlier, which again justifies its usage. We absolutely agree that a detailed evaluation of the ocean component would be a valuable addition to the presented more technical description of the model development. However, there are a couple of reasons why we decided to focus in this paper only on a base evaluation with respect to sea ice as the communicator between atmosphere and ocean (please take a look at the Author Comments to the Comments of Referee #1 for more details). However, as a reasonable compromise, because two of the three referees requested to include evaluation of the ocean component, we have incorporated a new subsection (section 3.5) in which the upper ocean temperature of HN1.2 and HN2.0 is compared with the PHC3.0 climatology. The upper ocean temperature was selected due to its direct influence on the sea-ice conditions. It also provides additional insight into the different bias structure of HN1.2 and HN2.0 and might be a valuable supplement to the discussion of model improvements that are directly related to the coupling. Because we have kept the focus mainly on a base evaluation with respect to sea ice as the communicator between atmosphere and ocean, and in order to clarify what the reader may (and may not) expect from the paper, we have added the following sentence to the abstract: "The evaluation focuses mainly on sea ice as the communicator between atmosphere and ocean." We have also added a brief outlook in the Conclusions section that more detailed evaluation of the model is postponed until the development process towards an improved configuration will have been completed and will be subject of follow-up studies.

C3: *More details on the HN1.2 runs are needed for a clearer understanding of the intercomparison results*

A3: A considerable number of details on the previous version of the coupled model has been added to the paper, whenever differences between HN2.0 and HN1.2 appeared to be helpful for understanding the intercomparison results.

C4: *The manuscript would largely benefit from a detailed analysis of the possible changes in the air-ocean-sea ice interactions*

A4: The decision trying to publish the upgraded version of the coupled model in GMD as "Development and technical paper" has been made in order to allow a more detailed description of technical details from which in particular model users may benefit, but also other model developers who want to couple different model components. Detailed analysis of the possible changes in the air-ocean-sea ice interactions would have a geoscientific focus and goes beyond the scope of a development and technical paper in GMD.

Response to the minor comments

In the following, a point-by-point response to the referee's minor comments is given in the sequence of comment (C) and answer (A).

C: *Pg 4, line 4: 0.25degree corresponds to approximately 27km at the equator. Is this the nominal resolution of the atmospheric grid?*

A: The horizontal resolution of 0.25° corresponds actually to approximately 27 km in the rotated spherical coordinate system where the equator crosses the model domain. The use of a rotated spherical coordinate system together with the location of its North Pole as only necessary information for its definition was already mentioned at the beginning of this section.

C: *Pg 5, line 10: provide a clearer distinction between the setting of stand-alone mode and the coupled mode*

A: Information on the HIRHAM5 stand-alone mode is not absolutely necessary and has therefore been removed in this paragraph in order to ease the understanding of the coupled mode which is the key aspect of the description.

C: *Pg 5, lines 11-12: what are the implications of different conditions applied to uncovered sea points?*

A: Because there are only a few sea grid points in HIRHAM5's boundary zone that are not covered by NAOSIM, and because these few sea grid points are far away from the NAOSIM domain (see Figure 1), there are no implications like discontinuities or the like. The different surface conditions are therefore of no importance for the coupling with NAOSIM.

C: *Pg 5: I suggest to add details on relevant differences between HRM and FRM*

A: The few relevant differences between HRM and FRM as used in the coupled model have been specified now.

C: *Pg 5: more details on the ocean components are needed, parameterizations, mesh, bathymetry, etc.*

A: We have added information on the tracer advection scheme and the parameterizations of friction and diffusion to section 2.2.1 and on the bottom topography (bathymetry) to section 2.2.4. The grid (mesh) was already described in every detail.

C: *Pg 7, lines 1-2: the description of Arakawa B grid corresponds to the description of C grid in 2.1.2*

A: The description of the Arakawa B-grid is correct, but we have now indicated explicitly that both horizontal velocity components are staggered in the same way, which is in contrast to the Arakawa C-grid. The description of the Arakawa C-grid in the previous section 2.1.2 (now section 2.1.3) was not incorrect, but incomplete, because we missed to mention the discrete staggering of $u$ in $x$-direction and of $v$ in and $y$-direction. We have added corresponding information to section 2.1.3.

C: *Pg 7, lines 15-19: a more accurate description of numerical instability is needed. How are the authors sure that the model crashes are all ascribed to the choice of the mixing time step?*

A: The reasons for numerical instability are hard to determine. At least the model crashes can definitely not be ascribed to the choice of the mixing time step, as the

referee supposes; they must have any other causes. We suppose that sharp gradients in the atmospheric fluxes, which often occur in the vicinity of coastal mountains, might trigger small-scale instabilities in the nearshore currents that cannot be resolved numerically. However, this is guesswork and should not be stated as explanation without any proof. Therefore, we did not state potential reasons for the occurrence of the numerical instability, but only an option how to overcome the model crash when it occurs. The choice of the mixing time step is such an option, since mixing time steps may damp time splitting characteristic of schemes centered in time. The Euler backward time step, which replaces the centered leapfrog time step at regular intervals, is beyond that diffusive and also damps spatial scales. Therefore, a smaller number of mixing time steps, which is in turn equivalent to a higher frequency of Euler backward time steps, was chosen in case of a model crash and has proved to be effective to overcome the crash. We have now indicated explicitly in the corresponding paragraph that the choice of a smaller number (of mixing time steps) is a suitable "option" to avoid a model crash, but we have abstained from adding any speculative assumptions on potential reasons for the occurrence of the model crash.

C: *Pg 7, lines 27-29: as for the atmospheric component, provide a clear distinction between stand-alone and coupled setting, and comment the possible implication of different forcing in coupled and uncoupled sectors of the domain*

A: Just as for the atmosphere component, information on the stand-alone mode has been removed also in this paragraph, since the coupled mode is the key aspect of the description. In contrast to the atmosphere component, the coupled and uncoupled sectors of the model domain adjoin each other, and it cannot be ruled out that discontinuities occur due to the different treatment of surface conditions. This is indeed the case in HN1.2 and earlier versions, where the atmospheric surface fluxes inside and outside the coupling domain systematically differ. In HN2.0, the atmospheric fluxes inside and outside the coupling domain basically agree in their temporal and spatial variation. Consequently, the transition from inside to outside the coupling domain is

rather smooth and the discontinuity does not appear any longer. It is suggested that this additional improvement in HN2.0 can not only be attributed to improved physical parameterizations in the atmosphere component HIRHAM5, but also to the new coupling procedure which applies conservative remapping and consistent time averaging of coupling fields with correct timing. We have added the discussion of the discontinuity at the boundary of the coupling domain to the new section 3.5, subsequent to the discussion of the upper ocean temperatures, and have recapitulated this additional improvement in HN2.0 again in the Conclusions section.

C: *Pg 8, lines 11-15: add a description on the new parallelization since this is one of the major improvements to the system. Which method has been used? Which component of the system is affected? Only NAOSIM is mentioned. How the impact on stand-alone simulations compares with coupled simulations?*

A: The paragraph has been extended by more specific information on the new parallelization of NAOSIM. Specifically, information on the parallelization method as well as on the number of processors allocated to HIRHAM5 and NAOSIM for coupled simulations and the corresponding elapsed time required to simulate one calendar year have been added. Since HIRHAM5 already comprised an efficient parallelization, there was no need for improvement in HIRHAM5 concerning this matter. Therefore, only NAOSIM is mentioned.

C: *Pg 8, line 16: motivate the choice of YAC version 1.2, also compared to the previous coupler.*

A: A brief description of the coupling procedure in HN1.2 and earlier versions has been added to section 2.3, directly leading to the motivation for using YAC in HN2.0.

C: *Pg 10, line 10: the namelist of ocean and sea ice parameters is missing in the manuscript*

A: Relevant namelist parameters (and other physical constants) of all model components are listed in Table B1. This is mentioned a few lines later at the end of the paragraph (previously page 11, lines 4–5).

C: *Pg 11, lines 6-14: this is unclear, I suggest to rewrite the entire paragraph adding precise information on the spin-up time for NAOSIM and HIRHAM, and for the coupled HN2.0 runs. The HN1.2 runs follow the same spin-up strategy? "The simulations were driven by ERAI data" refers to the coupled runs? If so, I suggest to reword, "driven" generally is for an ocean-sea ice simulation forced by atmospheric reanalysis.*

A: The paragraph already includes precise information on the initialization and the corresponding spin-up time of the HN2.0 ensemble simulations. The basic spin-up strategy is to initialize the ocean and sea-ice fields with different conditions from the steady state of the coupled spin-up run. The essential point is that the coupled spin-up run has to be carried out with the identical model version as used for the ensemble simulations themselves. This is necessary to avoid an initial model drift in the ensemble simulations due to differences in the model physics. From previous studies it is known that the coupled regional model needs a spin-up time of about 6–10 years to reach a quasi-stationary cyclic state of equilibrium (Dorn et al., 2007). If the initial ice conditions are not far away from this state, the spin-up time will be even shorter. This result was found in simulations with HN1.1 and has been experimentally verified with HN2.0. Since the initial conditions of all ensemble simulations with both HN2.0 and HN1.2 were taken after more than 10 years of spin-up, they all represent the steady state of the respective model version. Consequently, the HN1.2 ensemble uses the same spin-up strategy as the HN2.0 ensemble. The specific length of spin-up that exceeds the 10-year limit is completely irrelevant for this purpose. Clear information that all HN2.0 ensemble members were initialized with ocean–ice conditions from the steady state of the specific model configuration has now been added to the description of the ensemble simulation setup. Later, when we mention the HN1.2 ensemble for the first time, we have now pointed explicitly to the comparability of the two ensemble setups from a scientific point of view, even if they differ technically. Finally, the word "driven"

is commonly used in the regional model community for the lateral boundary forcing as well as for the lower or upper boundary forcing when applicable.

C: *Pg 11, line 20: how robust is the validation against ERA Interim since it has been used for the initialization? Why not to use independent data?*

A: The specific initialization of the atmosphere is only relevant in terms of numerical weather prediction; it absolutely plays no role in long-term climate simulations. Validation of the model simulations against ERAI data is therefore unproblematic. With respect to the lateral boundary forcing with ERAI data, it is even an advantage to validate the model simulations against ERAI data, because the effects of the internal model physics can better be isolated from large-scale atmospheric changes entering the model via the atmospheric model boundaries.

C: *Pg 11, line 22: reword "quasi realistic"*

A: The term "quasi realistic" has been replaced by "quite realistic".

C: *Pg 11, lines 27-28: clear statements on the differences and similarities between HN2.0 and HN1.2 would help, in addition to Graham et al 2017. I do not understand the sentence "Differences in the simulation results . . . indicate changes in the model performance". There are many differences between the 2 versions and probably between model set-up and spin-up. Please clarify.*

A: As aforementioned, the ensemble setup of HN1.2 and HN2.0 is comparable from a scientific point of view, even if it differs technically. Therefore the differences in the simulation results between the ensembles of HN1.2 and HN2.0 can surely be rated as indication of changes in the model performance due to differences in the physical characteristics, which includes not only physical parameterizations, but also the physical interaction between the model components by means of the revised coupling. We have added the reference of the scientific comparability of the two ensemble setups before pointing to the changes in the model performance. We have also specified

"model performance" as "model performance due to differences in the physical process descriptions" in order to express that we here do not refer to the technical or computational performance, which is clearly better in HN2.0, but is not expressed by the differences in the simulation results.

C: *Pg 12, lines 1-3: a good representation of sea ice properties does not guarantee a good representation of ocean and atmospheric fields. I think that assessing the quality of ocean/ atmosphere components would largely improve the manuscript. For example, how does the increased ocean resolution impact the ocean circulation and water properties in the Arctic and consequently the sea ice?*

A: Of course, a good representation of sea-ice properties does not guarantee a good representation of ocean and atmospheric fields. There might be model biases that are related for instance to shortcomings in the parameterization of clouds or boundary layer processes. On the other hand, a bad representation of sea ice properties inevitably induces biases in ocean and atmospheric fields due to biases in the surface energy budget. Sea ice thus plays a key role for the surface processes in the Arctic. A reasonable representation of sea ice in a coupled Arctic model system can be considered as a necessary, but not sufficient condition for a good representation of ocean and atmospheric fields. Atmospheric temperature fields are already discussed in the paper. We have extended the paper by a new subsection (section 3.5) in which the upper ocean temperature of HN1.2 and HN2.0 is compared with the PHC3.0 climatology. The upper ocean temperature was selected due to its direct linkage to the sea-ice conditions. It also provides additional insight into the different bias structure of HN1.2 and HN2.0 and might be a valuable supplement to the discussion of model improvements that are directly related to the coupling. The two component models HIRHAM5 and NAOSIM have already been used in stand-alone mode in previous studies that also include their evaluation. For example, HIRHAM5's cloud parameterization was evaluated by Klaus et al. (2012, 2016), and the impact of NAOSIM's increased resolution on the ocean circulation was investigated by Fieg et al. (2010). Since the setups of the component models have been left unchanged for the present coupled model version as far as possible, the primary task is to demonstrate that the new coupling procedure with the aid of YAC is technically working properly and that interactions between the component models are actually represented in an acceptable way. The latter is reflected in a good representation of sea ice.

C: *Pg 12, Figure 3. I suggest to compute the seasonal cycle over the same period for the three products.*

A: All data sets of the new Figure 3 refer now to the period 1979–2014. There are only minor, almost undetectable variances compared to the old figures so that adaptations of the text (beyond the figure captions) have not been needed.

C: *Pg 13, lines 4-5: where the thicker sea ice in HN2.0 comes from? The amplitude of the melting season is similar from the area/extent seasonal cycle. How different are the sea ice properties (concentration, thickness, temperature, etc.) in the initialization fields? Explain the different amplitude of the volume seasonal cycle between the 2 model versions.*

A: The amplitudes in the seasonal cycle of ice area and extent differ considerably (see Figure 3). The initial fields are completely irrelevant as evidenced by the low and completely insignificant across-ensemble scatter. Different amplitudes in the seasonal cycle of the ice volume are a consequence of different physical parameterizations in the two model versions as verified by a couple of sensitivity experiments (see Dorn et al., 2007, 2009).

C: *Pg 13, line 7: please define "relatively thick and thin ice". Maybe a distinction between pack ice and marginal zone ice may help. Which mechanisms (dynamics, thermodynamics, both) improve the thickness representation in HN2.0? Is the ice drift similar in the two models?*

A: We have replaced the term "relatively thick and thin ice" by the specification "rather

thick and rather thin ice within the Arctic pack ice region". Generally, thermodynamics plays the decisive role in terms of total ice volume/extent/area, while dynamics are jointly responsible for the geographical distribution of ice. The improved geographical distribution of ice can therefore also be attributed to an improved ice drift. The evaluation of sea-ice drift in HN2.0 will be subject of a follow-up study, which is currently in preparation. We have added a brief outlook in the Conclusions section that more detailed evaluation of the model will be subject of follow-up studies.

C: *Pg 13, line 9: change Januar to January*

A: "January" has been corrected.

C: *Pg 14, lines 5-6: could this differences in the growing season also be related to differences in the ocean and atmosphere between the two models? Are similar are, for example, air temperature and sea surface temperature in HN1.2 and HN2.0? Which one is the main driver of ice growth in the model?*

A: The newly incorporated Figure 7, which shows the upper ocean temperatures, clearly indicates colder ocean temperatures in the Kara and Chukchi seas in HN2.0. The early ice growth in these particular regions is certainly associated with the cold temperature bias in the upper ocean. This cold ocean temperature bias might be one of the reasons for accelerated ice growth and the overestimate of ice area and ice extent until the end of the year. We have added corresponding statements to the paragraph. HN2.0 also shows colder near-surface air temperatures in the afore-said regions, but they usually represent rather a response to the surface conditions than a driver. At the end of the paragraph, we have added some findings from a previous study (Dorn et al., 2007) about the linkage between near-surface air temperatures and ice growth as a function of the reference thickness for lateral freezing in order to explain why fine-tuning of this parameter represents one possibility to minimize the sea-ice bias as well as the winter temperature bias.

C: *Pg 14, line 12: the agreement between PIOMAS and HN2.0 is "reasonable" only*

*in March, the variability in September is not captured in the most recent years. For instance, the 2007 and 2012 minima are not reproduced. Given the modelled trend and the variability, I would not call "agreement" the overlap between curves. Why does sea ice extent in NH2.0 (mainly in March) present weaker inter-annual variability?*

A: We have replaced "agreement" by terms like "close to" or "similarities". The inter-annual variability in March sea-ice extent in NH2.0 is as weak as in the satellite data. HN1.2 shows here definitely too high interannual variability due to the Labrador Sea bias which spreads far to the south from time to time. In September, the interannual variability is in the order of magnitude as in the satellite data when looking at individual ensemble members. The slightly weaker variability of the ensemble mean results from the averaging process.

C: *Pg 14, line 16: "... trend in sea-ice volume ... can thus only arise from large-scale atmospheric changes". I do not believe that is true. What are the differences in the two oceans? Is the variability of air temperature the same in the models? How are the feedbacks between the two components affected by the new coupler?*

A: The only time-varying external forcing of the coupled model system is the atmospheric boundary forcing. All other climate forcings are given by fixed constants (like greenhouse gases, aerosols, ozone, solar constant, orbital parameters, land surface cover, soil characteristics) or constant seasonal cycles (like lateral ocean boundary, deep soil temperature, vegetation). In view of the fact that only the atmospheric boundary forcing changes with time and feedback cycles are internal to the climate system, there is no other explanation for the downward trend as the one given in the paper, regardless whether one believes it or not.

C: *Pg 14, lines 29-34: I did not understand the message within those lines. Please, rephrase.*

A: We have removed the last two sentences from this paragraph and have written instead: "Deviations from observed sea-ice conditions in specific years can therefore

also be a consequence of internal model variability." The main message is hereby better emphasized.

C: *Pg 16, line 5: Kelvin in the text and Celsius in Fig.7. Use the same.*

A: We have consistently replaced Kelvin by degrees Celsius.

C: *Pg 17, line 5: 0 degree is the freezing temperature of freshwater (no salt in it). This is not the case for the Arctic ocean upper layer. Rephrase.*

A: There is no salt in the ice at the top surface. $0\,°C$ is absolutely correct.

C: *Pg 17, line 7-8: about the reason of different summer temperature, how different are the heat fluxes between ocean and atmosphere in the two models? Is the air/ocean poleward heat transport the same? Then, rephrase "with an approximately by 10% underestimated sea ice concentration..."*

A: There are differences which are here beside the point. The clause in question has been reworded and reads now "... with the underestimation of sea-ice concentration by about $10\,\%$ ..."

C: *Pg 18, Figure 7: it might be more useful for the reader to have directly the plots of the differences HN2.0 – HN1.2 and HN2.0 – ERA Interim. It would help to add a contour indicating to the Arctic water freezing point.*

A: We followed the suggestion of Referee #1 and have replaced the subfigures of the HN1.2 and HN2.0 climatologies by their respective difference to ERA-Interim. The freezing temperature of sea water is not constant, but depends on the salinity. A single contour cannot be added. Also, it would not help understanding the differences.

C: *Pg 18, line 2: does the sea ice model include a melt-pond scheme? If so, which one? Was the same in HN1.2?*

A: The sea-ice model does not include a melt-pond scheme; but the atmosphere model applies the sea-ice albedo scheme of Køltzow (2007) which includes the effect of melt ponds as mentioned in section 2.1.2 (previously section 2.1.1). The same scheme was used in HN1.2. A detailed description of the scheme can be found in the reference paper of HN1.2 (Dorn et al., 2009).

C: *Pg 18: maybe a comment on differences in solid and liquid precipitation between the 2 models and the comparison with ERA Interim might be helpful*

A: An evaluation of precipitation in the model goes far beyond the scope of the current paper. Also, reanalysis products vary drastically in precipitation estimates over the Arctic Ocean (see Boisvert et al., 2018).

C: *Title of section 4 "Conclusions": I do not detect so many conclusions or discussion on the model performances, more future work. It would be nice to add some conclusions drawn from the model results; alternatively rename Section 4 to "Conclusions and Future work".*

A: We have added a few conclusions with respect to the performance of the revised coupling procedure and have changed the title of section 4 to "Conclusions and outlook".

C: *Pg 19, line 11-17: this study might also suggest that the physical core of the regional model components needs larger improvements. From those lines, a question arises whether the manuscript should include a better tuning and so better results. I would suggest to reformulate.*

A: There is no model that does not need improvements anymore. Arriving at model improvements is a long way that one can go only step by step. We think that publication of intermediate steps of model developments are of value as well. The indication of model weaknesses or even model configuration errors might be a poor selling point, but is an honest way that complies with the rules of good scientific practice.

C: *Pg 20, line 3: how are the snow and ice albedo defined in HN1.2?*

A: The definition of the snow and ice albedo in HN1.2 was detailed in the reference

paper of HN1.2 (Dorn et al., 2009). This has already been mentioned in section 2.1.2 (previously section 2.1.1). The few modifications in HN2.0 have been described in the same section.

C: *Pg 20, Code availability: add the link to Max Planck Institute webpage on YAC*

A: Information about access to the coupling software YAC has been added to the section 'Code availability'.

C: *I do not think that Table A1 and table A2 are necessary. For example, for the ocean depth, it might be enough adding something like: the layer thickness is 10m from the surface to 215m and then increases up to about 350m at the bottom.*

A: Tables A1 and A2 are not absolutely necessary, but helpful for model users, since this paper is intended as reference paper of HN2.0.

**References**

Adams, S., Willmes, S., Heinemann, G., Rozman, P., Timmermann, R., and Schröder, D.: Evaluation of simulated sea-ice concentrations from sea-ice/ocean models using satellite data and polynya classification methods, Polar Research, 30, 7124, doi:10.3402/polar.v30i0.7124, 2011.

Aksenov, Y., Karcher, M., Proshutinsky, A., Gerdes, R., de Cuevas, B., Golubeva, E., Kauker, F., Nguyen, A. T., Platov, G. A., Wadley, M., Watanabe, E., Coward, A. C., and Nurser, A. J. G.: Arctic pathways of Pacific Water: Arctic Ocean Model Intercomparison experiments, J. Geophys. Res. Oceans, 121, 27–59, doi:10.1002/2015JC011299, 2016.

Boisvert, L. N., Webster, M. A., Petty, A. A., Markus, T., Bromwich, D. H., and Cullather, R. I.: Intercomparison of precipitation estimates over the Arctic Ocean and its peripheral seas from reanalyses, J. Clim., 31, 8441–8462, doi:10.1175/JCLI-D-18-0125.1, 2018.

Dorn, W., Dethloff, K., Rinke, A., Frickenhaus, S., Gerdes, R., Karcher, M., and Kauker, F.: Sensitivities and uncertainties in a coupled regional atmosphere–ocean–ice model with respect

to the simulation of Arctic sea ice, J. Geophys. Res., 112, D10118, doi:10.1029/2006JD007814, 2007.

Dorn, W., Dethloff, K., and Rinke, A.: Improved simulation of feedbacks between atmosphere and sea ice over the Arctic Ocean in a coupled regional climate model, Ocean Model., 29, 103–114, doi:10.1016/j.ocemod.2009.03.010, 2009.

Drange, H., Gerdes, R., Gao, Y., Karcher, M., Kauker, F., and Bentsen, M.: Ocean General Circulation Modelling of the Nordic Seas, in: The Nordic Seas: An Integrated Perspective, edited by Drange, H., Dokken, T., Furevik, T., Gerdes, R., and Berger, W., vol. 158 of *Geophysical Monograph Series*, pp. 199–220, American Geophysical Union, Washington DC, doi:10.1029/158GM14, 2005.

Fieg, K., Gerdes, R., Fahrbach, E., Beszczynska-Möller, A., and Schauer, U.: Simulation of oceanic volume transports through Fram Strait 1995–2005, Ocean Dyn., 60, 491–502, doi:10.1007/s10236-010-0263-9, 2010.

Karcher, M., Smith, J. N., Kauker, F., Gerdes, R., and William M. Smethie, J.: Recent changes in Arctic Ocean circulation revealed by iodine-129 observations and modeling, J. Geophys. Res., 117, C08007, doi:10.1029/2011JC007513, 2012.

Klaus, D., Dorn, W., Dethloff, K., Rinke, A., and Mielke, M.: Evaluation of two cloud parameterizations and their possible adaptation to Arctic climate conditions, Atmosphere, 3, 419–450, doi:10.3390/atmos3030419, 2012.

Klaus, D., Dethloff, K., Dorn, W., Rinke, A., and Wu, D. L.: New insight of Arctic cloud parameterization from regional climate model simulations, satellite-based, and drifting station data, Geophys. Res. Lett., 43, 5450–5459, doi:10.1002/2015GL067530, 2016.

Køltzow, M.: The effect of a new snow and sea ice albedo scheme on regional climate model simulations, J. Geophys. Res., 112, D07110, doi:10.1029/2006JD007693, 2007.

Kriegsmann, A. and Brümmer, B.: Cyclone impact on sea ice in the central Arctic Ocean: a statistical study, The Cryosphere, 8, 303–317, doi:10.5194/tc-8-303-2014, 2014.

Rabe, B., Karcher, M., Kauker, F., Schauer, U., Toole, J. M., Krishfield, R. A., Pisarev, S., Kikuchi, T., and Su, J.: Arctic Ocean basin liquid freshwater storage trend 1992–2012, Geophys. Res. Lett., 41, 961–968, 10.1002/2013GL058121, 2014.

Rozman, P., Hölemann, J. A., Krumpen, T., Gerdes, R., Köberle, C., Lavergne, T., Adams, S., and Girard-Ardhuin, F.: Validating satellite derived and modelled sea-ice drift in the Laptev Sea with in situ measurements from the winter of 2007/08, Polar Research, 30, 7218, doi:10.3402/polar.v30i0.7218, 2011.

---

## Author Comment (AC3) · 18 Mar 2019

**Author Comments to the Comments of Referee #3**

Response to the general comments

In the following, a point-by-point response to the referee's general comments is given in the sequence of comment (C#) and answer (A#), where # refers to the numbering of the referee's general comments.

[Figure]

C1: *Since the main purpose of this paper is to document changes to the coupled model, it is important that this description be clear and thorough. However, the authors fail to clearly present the differences between the new version HN2.0 and the previous version (HN1.x). The main change is the use of a new atmospheric model, which itself is built on two previsouly described components (HIRLAM7 and ECHAM5.4). The model description is often quite difficult to follow as the authors intermingle modifications with respect to HN1.x with modifications to HIRLAM7 and ECHAM5.4. I recommend this section be rewritten to make these differences clear. In particular, if the aim here is to document the differences between HN2.0 and HN1.2 than these should be outlined in detail, and not rely on previous publications of HIRLAM7 and ECHAM5.4. Without a clear description of direct differences between HN1.2 and HN2.0 it is difficult to interpret the results of the model evaluation presented in Section 3.*

A1: We agree that the description of the changes and modifications was not always clear. In particular, a clear differentiation between the general HIRHAM5 description and specific modifications directly related to HN2.0 was missing in the description of the atmosphere model. In the revised version of the paper, we have specified the technical modifications in HIRHAM5 compared to HIRLAM7 and ECHAM5 in a separate section. The subsequent section provides a brief overview about the physical parameterizations and general differences to HN1.x followed by the specific modifications in HN2.0. In addition, the lead paragraph of section 2 includes now basic differences between HN2.0 and HN1.x in order to avoid any misunderstanding from the outset. This includes information on the model resolutions in HN2.0 and HN1.x which were previously missing for HN1.x. Further differences between HN2.0 and HN1.x are explicitly stated when describing the model components.

C2: *While I agree that sea ice is an important indicator of overall model performance, a reference paper such as this is more useful when a broader presentation of model performance is outlined. Given the large changes in the atmospheric component I would have expected to see a more detailed description of characteristics of the modelled*

[Figure]

*atmosphere.*

A2: The regional atmospheric climate model HIRHAM5 was developed a few years ago and has already been used in a number of previous studies that partly include very detailed description of characteristics of the modeled atmosphere. We have added citations to some of these previous studies to the preface of section 2.1, primarily to avoid the impression that the development of HIRHAM5 is related to the development of HN2.0. This is not the case; HIRHAM5 already existed before and was chosen as the new atmosphere component of the coupled system, precisely because it has already successfully been applied in a number of previous studies. The aim of the present paper is to document the new coupled system, and not the individual components, and to demonstrate that the new coupling procedure with the aid of YAC is technically working properly and that interactions between the component models are actually represented in an acceptable way. Sea ice is the communicator between atmosphere and ocean. Its reasonable representation in HN2.0 clearly indicates that the coupling works well, even if there is still need for further improvements in the model configuration. A full evaluation of the entire model would be so substantial that one or more stand-alone papers are required or at least highly recommended. Apart from this, a detailed evaluation of different aspects of the Arctic climate system would have a geoscientific focus and goes beyond the scope of a development and technical paper in GMD. Some of these aspects, for instance sea-ice drift, Atlantic water inflow, and atmospheric cyclones, are already subject of our current research and will likely result in follow-up papers in pure scientific journals.

Response to the specific comments

In the following, a point-by-point response to the referee's specific comments is given in the sequence of comment (C#) and answer (A#), where # refers to the numbering of the referee's specific comments.

C1: *Pg1, line 10: "allow to simulate". Please rephrase, perhaps "allow one to simulate" or similar.*

A1: We have replaced "allow to simulate" by "provide the possibility to simulate".

C2: *Section 1, para 3: It would be helpful to explain the motivations for upgrading the atmospheric component and any particular deficiencies that it is aiming to overcome. Also, the choice for the particular components chose for HN2.0 could be justified (ie HIRLAM7 and ECHAM5).*

A2: The motivation for upgrading the atmospheric component is simply to make use of the most recent HIRHAM version which includes more sophisticated parameterizations. This HIRHAM version, named HIRHAM5, was built up with HIRLAM7 and ECHAM5 a few years ago and has already been used in a number of previous studies (and a subset of them has now explicitly been cited). The development of HIRHAM5 is not related to the development of HN2.0, it is just the atmospheric component of the new coupled system. We have realized that there is need for a clear differentiation between the general HIRHAM5 description and specific modifications directly related to HN2.0, and we have revised section 2.1 accordingly.

C3: *Pg2, line 18: Regardless if they have been described in reference manuals, if the aim of this paper is to document the new model version than a description of model components should be provided here.*

A3: The two model components are not part of the development of the coupled system. Both HIRHAM5 and NAOSIM were developed a few years ago and have already been used in previous studies that partly also include more detailed model descriptions. The aim of the present paper is to document the new coupled system and not the individual components, although we agree that it could be helpful to go into more detail when describing the components. In the revised version of the paper, we have added more information on the components, but we have kept the focus on specific changes for the coupled model system.

C4: *Pg. 3, line 6: spelling error, should be "aerosol"*

A4: Should actually be "aerosols". Corrected!

C5: *Pg. 3, line 8-9: "The most important modification" from what? From HN1.2 or ECHAM5? Mixing these up makes the text difficult to follow. Also statements like "for the most part" are vague and should be avoided. Rather, explain what has been changed and what hasn't.*

A5: All modifications refer to the ECHAM5 parameterizations as component of HIRHAM5. HN1.2 comprises the atmosphere model HIRHAM4, a different model, except for the name HIRHAM. This circumstance has now explicitly been mentioned. Further, we have restructured the division into subsections to differentiate between the general descriptions of HIRHAM5 (with previous modifications with respect to the original HIRLAM-7.0 and ECHAM5 codes) and the current modifications for HN2.0. In the revised version of the paper, there is now a subsection for the HIRHAM5 components with general technical modifications compared to the original model codes and a subsection for modified parameterizations as part of the development of HN2.0. Ambiguous formulations have been reworded. The statement "for the most part" has been removed.

C6: *Pg. 3, line 17- 18: "...attenuated such that at least 25%...". This sentence is quite difficult to follow. If this is the most important modification then it would be worth including the equation and describing this properly. Also it seems it may be relevant for the sea ice results presented in Section 3 (?).*

A6: If we included only the equation for the restriction of the melt pond fraction, we would explicitly emphasize a non-observationally based model adjustment. A reader of the paper might think that the equation with a value of 25 % represents a general improvement of the albedo parameterization, just because the equation is specified. This is not the case. We already noted in the manuscript that the value of 25 % can be considered as a tuning value and that a more realistic parameterization of the fractions

of snow and melt ponds should be derived from observations. First efforts towards an observationally based parameterization of the melt pond fraction are already underway. And, of course, the albedo parameterization is highly relevant for sea ice as is generally known.

C7: *Pg. 3, Line 22: "The second modification. . . ". From what? ECHAM or HN1.2?*

A7: All modifications refer to HIRHAM5 as noted before.

C8: *Pg. 4, line 13: semi-Lagrangian advection schemes are known to have conservation issues when used at high CFL number. For a weather model this is usually not a problem, but for a regional climate model this could affect the results. A discussion of this issue and the extent to which HN2.0 is conservative should be included, perhaps with some demonstration of applicable CFL numbers.*

A8: Every numerical scheme has its pros and cons. Atmospheric climate models usually base on NWP models and need to rely on their skill in simulating the weather using the specifically implemented numerical scheme. The reasonable simulation of the frequency of occurrence of dominant weather pattern is also essential for atmospheric climate models, certainly more important than everything else. HIRLAM was and is successfully applied by a number of European weather services, and also HIRHAM5 was and is applied by a couple of research institutions. So far, conservation issues using the semi-Lagrangian scheme with large time steps in HIRLAM or HIRHAM5 have never been documented in the literature. The time step of 600 s was chosen, because the model still runs stable with this time step, and produces results comparable to simulations using the Eulerian advection scheme, even if the CFL number is larger than one. Assuming that HIRLAM and HIRHAM5 does not have conservation issues, why should HN2.0 have them? We would understand if such a comment was addressed to authors of a HIRLAM or HIRHAM5 development paper. For the description of the coupled model system, the discussion of a non-existent issue in one of its components does not provide an added value.

C9: *Pg. 5, line 17: "fine resolution" and "high-resolution" are not very useful. Please in-clude a more precise indication of model resolution. Also, on pg2, line 16 it is noted that the ocean component is "largely the same". If the model configuration has completely changed this statement is not accurate. Moreover, simply stating that the difference in model configuration is described in Fieg et al (2010) is not sufficient. At least a brief description should be provided here as well.*

A9: We have added basic differences between HN2.0 and HN1.2 to the lead paragraph of section 2. This also includes information on the different model resolution. The phrase "largely the same" has been dropped in the course of reformulating the lead paragraph. Subsequent to the reference to Fieg et al. (2010), we have added a brief overview about the few differences between HN2.0 and HN1.2.

C10: *Section 2.2.3: How is this different from HN1.x?*

A10: The ocean-sea ice coupling in HN1.x and HN2.0 is identical. The optional scheme by Castellani et al. (2014) is not available in HN1.x, but is also not used in HN2.0.

C11: *Section 2.2.4: How is this different from HN1.x?*

A11: The model domain is the same, but HN1.x uses lower horizontal, vertical, and temporal resolution than HN2.0. We have now explicitly specified the resolutions of HN2.0 and HN1.x in the lead paragraph of section 2. Model crashes did not occur in HN1.x., but they did occur in stand-alone simulations with the FRM version of NAOSIM. Evidently, they are not related to the coupling with HIRHAM5.

C12: *Pg. 7, line 29: Is there any blending used when going from HIRHAM5 forcing to ERAI?*

A12: Blending is not used, neither in HN2.0 nor in HN1.2. Especially HN1.2 shows dis-continuities in the atmospheric surface fluxes inside and outside the coupling domain. In HN2.0, the transition from inside to outside the coupling domain is rather smooth

and the discontinuity does not appear any longer. We have added the discussion of the discontinuity at the boundary of the coupling domain to the new section 3.5, subsequent to the discussion of the upper ocean temperatures, and have emphasized this additional improvement in HN2.0 again in the conclusions.

C13: *Pg. 8, line 1: "standard bulk formulas". Please describe. Are these the same bulk formulas used by HIRLAM when coupled?*

A13: We have now specified that the bulk formulas used in NAOSIM are based on the formulations for turbulent fluxes and shortwave radiation by Parkinson and Washington (1979) and for longwave radiation by Rosati and Miyakoda (1988). Further, any bulk formulas of HIRLAM are completely irrelevant for HN2.0, since the originally embedded physical parameterization package of HIRLAM was replaced by that of ECHAM5 as already mentioned on page 2 of the manuscript. ECHAM5 includes bulk formulas only for turbulent surface fluxes. These formulas are more sophisticated, since they explicitly take account of atmospheric stability in the near-surface layer.

C14: *Pg. 8, line 8: Is this the only difference in how fluxes are calculated? For example, are surface roughnesses and boundary layer stability all treated the same?*

A14: There are numerous differences in the calculation of fluxes between sophisticated atmosphere models like HIRHAM5 and simple bulk formulas. However, from the viewpoint of the coupled model system, it is enough to know where the fluxes come from and not how they were calculated. There is no need to detail the differences in the calculation at this point.

C15: *Pg. 11, line 10: The use of ice-ocean fields from Januaries 1991 to 2000 seems a rather odd choice. Some explanation should be provided. Also, since thickness over this period were thinner than for the earlier period, please describe any impact on mean sea ice results (i.e. due you see any spin up effects? Is there any change is ensemble spread from year1 to year 20+?*

A15: The choice of initial ocean and sea-ice fields from different years of a spin-up run is motivated by the fact that the real ice-ocean state in January 1979 is practically unknown. The different initial conditions from the spin-up run represent the diversity of ocean–ice conditions within the steady state of the specific model configuration. Two essential points are that the spin-up run already reached a quasi-stationary seasonal-cyclic state of equilibrium for the mid-1980s and that the spin-up run was carried out with the identical model version as used for the ensemble simulations themselves. The latter is necessary to avoid an initial drift in the ensemble simulations due to inconsistent model physics. Consequently, spin-up effects in the ensemble simulations are negligible. Systematic temporal changes in the ensemble spread are not visible, too. The specific method for choosing initial conditions is well-conceived and was already applied in previous studies (e.g., Dorn et al., 2012). Clear information that all ensemble members were initialized with ocean and sea-ice fields that represent the diversity of ocean–ice conditions within the steady state of the specific model configuration has now been added to the description of the ensemble simulation setup.

C16: *Pg. 11, line 27: If the main comparison presented in this paper is against this HN1.2 ensemble, than an explanation of how the setup differs should be given.*

A16: We have replaced "differs slightly" by "differs technically", because the differences in the two ensemble setups are only of technical nature. Further, we have specified now that the initial ocean and sea-ice fields were taken from different years of two earlier simulations with HN1.2, and have emphasized that the two ensemble setups are comparable from a scientific point of view.

C17: *Fig. 3: (middle). It would be helpful to include PIOMAS here as well to be able to differentiate spatial differences (area) from thickness contributions to total volume.*

A17: Ice area/extent from PIOMAS differs slightly from the satellite data, especially during the summer months. The sea-ice differences between HN2.0 and PIOMAS are indeed lower than those between HN2.0 and the satellite data. Nevertheless, we think

that the satellite data are more reliable than the model assimilation system PIOMAS, and we decided then to not include ice area/extent from PIOMAS in Figure 3 (as well as in Figure 5), just to avoid confusion in the interpretation of the figure by the presence of two observational data sets. This argument still holds true.

C18: *Pg. 14, line 7: "had been resulted" change to ". . . resulted. . . " or similar.*

A18: Has been corrected to "had resulted".

C19: *Pg. 14, line 19: "observation-like" is a bit of an unusual term. Perhaps change this to "reference"*

A19: We have replaced "observational-like" by "reference".

C20: *Pg. 14, line 32-33: The inability to simulate extrema is not necessary just a matter of model internal variability though as many key processes are missing (e.g. wave-ice interactions which played an important role in the 2012 minimum that is used as an example). It would be good to note this limitation in simulating extremes and comment on the degree to which this may be important for simulations with this regional climate model. Since HN2.0 has a higher resolution ocean-ice model, does this affect extremes?*

A20: We agree completely that the inability to simulate extrema is not necessarily just a matter of model internal variability. The special note to sea-ice extrema by exemplifying the sea-ice minimum in 2012 has been removed from this paragraph. Instead, we have written: "Deviations from observed sea-ice conditions in specific years can therefore also be a consequence of internal model variability."

C21: *Pg. 17, line 5-10. It would be helpful to show some additional diagnostics here associated with albedo and surface heat fluxes to understand better the source of these differences.*

A21: In-depth analysis of the differences is topic of upcoming studies and goes beyond the scope of a development and technical paper in GMD.

C22: *Pg. 18, line 3: If there is increased melting from the ocean, may this also be related to changes in ocean transports. A higher resolution ocean configuration may allow more Atlantic water to enter the Arctic via Fram strait and the Barents Sea. Some comment/validation of this would be helpful to understand how the behavior of HN2.0 differs from HN1.2.*

A22: The ocean mixed layer beneath the Arctic sea ice is more or less decoupled from the Atlantic water due to the stable halocline in the Arctic Ocean that minimizes the vertical exchange. Except for the marginal and shelf seas of the Arctic Ocean, potential changes in ocean transports play only a minor role for bottom melting of sea ice. The heat content of the Arctic ocean mixed layer is primarily controlled by the heat exchange with the atmosphere. This applies to both HN2.0 and HN1.2. The impact of the different resolution in NAOSIM on the ocean circulation was investigated by Fieg et al. (2010) with a focus on the oceanic transports through Fram Strait.

C23: *Pg. 19, line2: "...be solved until now" change to "...as of now".*

A23: The term "as of now" has a different meaning than "until now". The latter is what we would like to emphasize at this point.

**References**

Castellani, G., Lüpkes, C., Hendricks, S., and Gerdes, R.: Variability of Arctic sea-ice topography and its impact on the atmospheric surface drag, J. Geophys. Res. Oceans, 119, 6743–6762, doi:10.1002/2013JC009712, 2014.

Dorn, W., Dethloff, K., and Rinke, A.: Limitations of a coupled regional climate model in the reproduction of the observed Arctic sea-ice retreat, The Cryosphere, 6, 985–998, doi:10.5194/tc-6-985-2012, 2012.

Fieg, K., Gerdes, R., Fahrbach, E., Beszczynska-Möller, A., and Schauer, U.: Simulation of oceanic volume transports through Fram Strait 1995–2005, Ocean Dyn., 60, 491–502, doi:10.1007/s10236-010-0263-9, 2010.

Parkinson, C. L. and Washington, W. M.: A large-scale numerical model of sea ice, J. Geophys. Res., 84, 311–337, 1979.

Rosati, A. and Miyakoda, K.: A general circulation model for upper ocean simulation, J. Phys. Oceanogr., 18, 1601–1626, 1988.